# Evaluating model performance under worst-case subpopulations

**Mike Li***
Decision, Risk, and Operations Division
Columbia Business School
New York, NY 10027
MLi24@gsb.columbia.edu

**Hongseok Namkoong**
Decision, Risk, and Operations Division
Columbia Business School
New York, NY 10027
namkoong@gsb.columbia.edu

**Shangzhou Xia**
Decision, Risk, and Operations Division
Columbia Business School
New York, NY 10027
SXia24@gsb.columbia.edu

## Abstract

The performance of ML models degrades when the training population is different from that seen under operation. Towards assessing distributional robustness, we study the worst-case performance of a model over *all* subpopulations of a given size, defined with respect to core attributes $Z$. This notion of robustness can consider arbitrary (continuous) attributes $Z$, and automatically accounts for complex intersectionality in disadvantaged groups. We develop a scalable yet principled two-stage estimation procedure that can evaluate the robustness of state-of-the-art models. We prove that our procedure enjoys several finite-sample convergence guarantees, including *dimension-free* convergence. Instead of overly conservative notions based on Rademacher complexities, our evaluation error depends on the dimension of $Z$ only through the out-of-sample error in estimating the performance conditional on $Z$. On real datasets, we demonstrate that our method certifies the robustness of a model and prevents deployment of unreliable models.

## 1 Introduction

The training population typically does not accurately represent what the model will encounter under operation. Model performance has been observed to substantially degrade under distribution shift [16, 28, 69, 80, 53] in speech recognition [52], automated essay scoring [4], and wildlife conservation [11]. Similar trends persist for state-of-the-art NLP and computer vision models [78, 74], even on new data constructed under a near-identical process [57, 66]. Heavily engineered commercial models are no exception [19], performing poorly on rare entities in named entity linking and examples that require abstraction and distillation in summarization tasks [38].

A particularly problematic form of distribution shift comes from embedded power structures in data collection. Data forms the infrastructure on which we build prediction models [30], and they inherit socioeconomic and political inequities against marginalized communities. For example, out of 10,000+ cancer clinical trials the National Cancer Institute funds, less than 5% of participants were non-white [21]. Typical models replicate and perpetuate such bias, and their performance drops significantly on underrepresented groups. Speech recognition systems work poorly for Blacks [52]

---

*Authors ordered alphabetically.

35th Conference on Neural Information Processing Systems (NeurIPS 2021).

and those with minority accents [3]. More generally, model performance degrades across demographic attributes such as race, gender, or age, in facial recognition, video captioning, language identification, and academic recommender systems [41, 46, 17, 72, 79, 19].

Model training typically relies on varied engineering practices. It is crucial to *rigorously certify* model robustness prior to deployment for these heuristic approaches to bear fruit and transform consequential applications. Ensuring that models perform uniformly well across subpopulations is simultaneously critical for reliability, fairness, satisfactory user experience, and long-term business goals. While a natural approach is to evaluate performance across a small set of groups, disadvantaged subpopulations are hard to define a priori because of *intersectionality*. The most adversely affected are often determined by a complex combination of variables such as race, income, and gender [19]. For example, performance on summarization tasks varies across demographic characteristics and document specific traits such as abstractiveness, distillation, and location and dispersion of information [38].

Motivated by these challenges, we study the worst-case subpopulation performance across *all* subpopulations of a given size. This conservative notion of performance evaluates robustness to unanticipated distribution shifts in $Z$, and automatically accounts for complex intersectionality by virtue of being agnostic to demographic groupings. Formally, let $Z$ be a set of core attributes that we wish to guarantee uniform performance over. It may include protected demographic variables such as race, gender, income, age, or task-specific information such as length of the prompt or metadata on the input; notably, it can contain any continuous or discrete variables. We let $X \in \mathcal{X}$ be the input / covariate, and $Y \in \mathcal{Y}$ be the label. In NLP and vision applications, $X$ is high-dimensional and typically $\dim(Z) \ll \dim(X)$.

We use $\theta(X)$ to denote a fixed prediction model and consider flexible and abstract losses $\ell(\theta(x); y)$. Our goal is to ensure that the model $\theta$ performs well over all subpopulations defined over $Z$. We evaluate model losses on a mixture component, which we call a subpopulation. Postulating a lower bound $\alpha \in (0, 1]$ on the demographic proportion (mixture weight), we consider the set of subpopulations of the data-generating distribution $P_Z$

$$\mathcal{Q}_\alpha := \{Q_Z \mid P_Z = aQ_Z + (1-a)Q'_Z \text{ for some } a \geq \alpha, \text{ and subpopulation } Q'_Z\}. \qquad (1)$$

The demographic proportion (mixture weight) $a$ represents how underrepresented the subpopulation is under the data-generating distribution $P_Z$. Before deploying the model $\theta$, we wish to evaluate the worst-case subpopulation performance

$$\mathsf{W}_\alpha(\theta) := \sup_{Q_Z \in \mathcal{Q}_\alpha} \mathbb{E}_{Z \sim Q_Z} \left[\mathbb{E}[\ell(\theta(X), Y) \mid Z]\right]. \qquad (2)$$

The worst-case subpopulation performance (2) guarantees uniform performance over subpopulations (1) and has a clear interpretation that can be communicated to diverse stakeholders. The minority proportion $\alpha$ can often be chosen from first principles, e.g., we wish to guarantee uniformly good performance over subpopulations comprising at least $\alpha = 20\%$ of the collected data. Alternatively, it is often informative to study the threshold level of $\alpha^\star$ when $\alpha \mapsto \mathsf{W}_\alpha(\theta)$ crosses the *maximum level of acceptable loss*. The threshold $\alpha^\star$ provides a *certificate of robustness* on the model $\theta(\cdot)$, guaranteeing that all subpopulations large than $\alpha^\star$ enjoy good performance.

We develop a principled and scalable procedure for estimating the worst-case subpopulation performance (2) and the certificate of robustness $\alpha^\star$. A key technical challenge is that for each data point, we observe the loss $\ell(\theta(X); Y)$ but never observe the conditional risk evaluated at the attribute $Z$

$$\mu(Z) := \mathbb{E}[\ell(\theta(X); Y) \mid Z]. \qquad (3)$$

In Section 2, we propose a two-stage estimation approach where we compute an estimate $\widehat{h}_1(\cdot)$ of the conditional risk $\mu(\cdot)$. Then, we compute a plug-in estimate of the worst-case subpopulation performance under $\widehat{h}_1(\cdot)$ using a dual reformulation of the worst-case problem (2). We show several theoretical guarantees for our estimator of the worst-case subpopulation performance (2). Our first finite-sample result (Section 3.1) shows convergence at the rate $O_p\left(\sqrt{\mathfrak{Comp}_n(\mathcal{H})/n}\right)$, where $\mathfrak{Comp}_n$ denotes a notion of complexity for the model class estimating the conditional risk (3).

In some applications, it may be natural to define $Z$ using images or natural languages describing the input and use deep networks to predict the conditional risk (3). As the complexity term $\mathfrak{Comp}_n(\mathcal{H})$ becomes prohibitively large in this case [10, 86], our second result (Section 3.2) shows data-dependent

*dimension-free* concentration of our two-stage estimator: our bound only depends on the complexity of the model class $\mathcal{H}$ through the out-of-sample error for estimating the conditional risk (3). This error can be made small using overparameterized deep networks, allowing us to estimate the conditional risk (3) using even the largest deep networks and still obtain a theoretically principled upper confidence bound on the worst-case subpopulation performance. Leveraging these guarantees, we develop principled procedures for estimating the certificates of robustness $\alpha^{\star}$ in Section 3.3.

In Section 4, we demonstrate the effectiveness of our procedure on real data. By evaluating model robustness under subpopulation shifts, our methods allow the selection of robust models before deployment as we illustrate using the recently proposed CLIP model [62].

**Related work.** The long line of works on distributionally robust optimization (DRO) aims to *train models* to perform well under distribution shifts. Previous approaches considered finite-dimensional worst-case regions such as constraint sets [29, 39, 5] and those based on notions of distances for probability measures such as $f$-divergences [12, 13, 56, 55, 60, 33, 32], Levy-Prokhorov [34], Wasserstein distances [35, 73, 15, 37, 14, 82], and integral probability metrics based on reproducing kernels [77, 87]. The distribution shifts considered in these approaches are often contrived and difficult to interpret and often result in overly conservative models. Furthermore, these approaches do not currently scale to modern large-scale NLP or vision applications.

Our work is most closely related to Duchi et al. [31], who proposed algorithms for *training* models with respect to the worst-case subpopulation performance (2), which is a more ambitious goal than our narrower viewpoint of *evaluating* model performance pre-deployment. Their (full-batch) training procedure requires solving a convex program with $n^2$ variables per gradient step, which is often prohibitively expensive. Furthermore, training with respect to the worst-case conditional risk $\mathbb{E}[\ell(\theta(X); Y) \mid Z]$ do not scale to deep networks that can overfit to the training data [70]. By contrast, our evaluation perspective aims to take advantage of the rapid progress in deep learning. We build scalable evaluation methods that apply to arbitrary models, which allows leveraging state-of-the-art engineered approaches for training $\theta(\cdot)$. Our narrower focus on evaluation allows us to provide convergence rates that scale advantageously with the dimension of $Z$, compared to the nonparametric $O_p(n^{-1/d})$ rates for training [31]. Recently, Jeong and Namkoong [48] studied a similar notion of worst-case subpopulation performance in causal inference.

Our notion of worst-case subpopulation performance is also related to the by now vast literature on fairness in ML. We give a necessarily abridged discussion and refer readers to Barocas et al. [8] and Corbett-Davies and Goel [27] for a comprehensive treatment. A large body of work studies *equalizing* a notion of performance over fixed, pre-defined demographic groups for *classification tasks* [24, 36, 7, 43, 51, 84]. Kearns et al. [49, 50], Hébert-Johnson et al. [45] consider finite subgroups defined by a structured class of functions over $Z$, and study methods of equalizing performance across them. By contrast, our approach instantiates Rawls' theory of distributive justice [64, 65], where we consider the allocation of the loss $\ell(\cdot; \cdot)$ as a resource. Rawls' difference principle maximizes the welfare of the worst-off group and provides incentives for groups to maintain the status quo [64]. Similarly, Hashimoto et al. [44] studied negative feedback loops generated by user retention—they use a more conservative notion of worst-case loss than ours—as poor performance on a currently underrepresented user group can have long-term consequences.

Our diagnostics complement the recent approaches to benchmarking under distribution shifts [80, 66, 74, 53, 71, 78, 57] as our procedure does not require out-of-distribution data. Since good performance on a particular distribution shift does not necessitate robustness, we evaluate models using the worst-case subpopulation performance (2).

## 2 Approach

We begin by contrasting our approach to standard alternatives that consider pre-defined, fixed demographic groups [59]. Identifying disadvantaged subgroups a priori is often challenging as they are determined by *intersections* of multiple demographic variables. To illustrate such complex intersectionality, consider a drug dosage prediction problem for Warfarin [26], a common anti-coagulant (blood thinner). Taking the best prediction model for the optimal dosage on this dataset based on genetic, demographic and clinical factors [26], we present the squared error on the root dosage. In Figure 1, when age and race are considered *simultaneously* instead of *separately*, subpopulation performance vary significantly across intersectional groups.

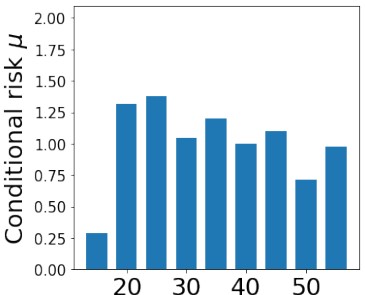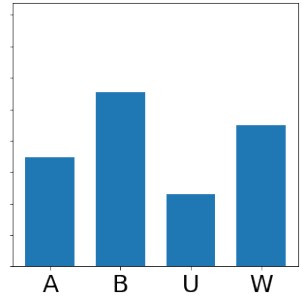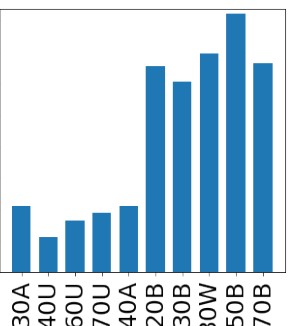

**Figure 1.** Conditional risk $\mu(Z) = \mathbb{E}[(Y - \theta(X))^2 \mid Z]$. Here $Z = age$ on the left panel, $Z = race$ in the center, and $Z = (age, race)$ on the right. A = Asian, B = Black, U = Unknown, W = White.

The worst-case subpopulation performance (2) automatically accounts for latent intersectionality. It is agnostic to demographic groupings and allows considering infinitely many subpopulations that represent at least $\alpha$-fraction of the training population $P$. By allowing the modeler to select arbitrary protected attributes $Z$, we are able to consider potentially complex subpopulations. For example, $Z$ can even be defined with respect to a natural language description of the input $X$. The choice of $Z$—and subsequent worst-case subpopulation performance (2) of the conditional risk $\mu(Z) = \mathbb{E}[\ell(\theta(X); Y) \mid Z]$—interpolates between the most conservative notion of subpopulations (when $Z = (X, Y)$) and simple counterparts defined over a single variable.

The choice of the subpopulation size $\alpha$ should be informed by domain knowledge—desired robustness of the system—and the dataset size relative to the complexity of $Z$. Often, proxy groups can be used for selecting $\alpha$. If we wish to ensure good performance over patients of all races aged 50 years or older, we can choose $\alpha$ to be the proportion of the least represented $(race, age \geq 50)$ group—this leads to $\alpha = 5\%$ in the Warfarin data. The corresponding worst-case subpopulation performance (2) guarantees good performance over all groups of similar size.

When it is challenging to commit to a specific subpopulation size, it may be natural to postulate a *maximum level of acceptable loss* $\bar{\ell}$. To measure the robustness of a model, we define the smallest subpopulation size $\alpha^\star(\theta)$ for which the worst-case subpopulation performance is acceptable

$$\alpha^\star(\theta) := \inf\{\alpha : \mathsf{W}_\alpha(\theta) \leq \bar{\ell}\}. \tag{4}$$

This provides a *certificate of robustness*: if $\alpha^\star(\theta)$ is large, then $\theta$ is brittle against even majority subpopulations; if it is sufficiently small, then $\theta$ performs well on underrepresented subpopulations.

We now derive estimators for the worst-case subpopulation performance (2) and the certificate of robustness (4), based on i.i.d. observations $(X_i, Y_i, Z_i)_{i=1}^n \sim P$. We assume our observations are independent from the data used to train the model $\theta(\cdot)$.

**Dual reformulation**  The worst-case subpopulation performance (2) is unwieldy as it involves an infinite dimensional optimization problem over probabilities. Instead, we use its dual reformulation for tractable estimation. We denote $[\cdot]_+ = \max(\cdot, 0)$, and abuse notation by letting $\mathsf{W}_\alpha(h)$ be the worst-case subpopulation performance for $h(Z)$ (so that $\mathsf{W}_\alpha(\theta) = \mathsf{W}_\alpha(\mu)$).

**Lemma 1** (Shapiro et al. [75, Example 6.19] and Rockafellar and Uryasev [67]). *If* $\mathbb{E}[h(Z)_+] < \infty$,

$$\mathsf{W}_\alpha(h) := \sup_{Q_Z \in \mathcal{Q}_\alpha} \mathbb{E}_{Z \sim Q_Z}[h(Z)] = \inf_{\eta \in \mathbb{R}} \left\{ \frac{1}{\alpha} \mathbb{E}_P[h(Z) - \eta]_+ + \eta \right\}. \tag{5}$$

The dual optimum is attained at the $(1 - \alpha)$-quantile of the $h(Z)$ [68, Theorem 10]. The dual (5) hence shows $\mathsf{W}_\alpha(\theta)$ is a tail-average of $\mu(Z)$, a popular risk measure known as the conditional value-at-risk (CVaR) in portfolio optimization [67].

---

**Algorithm 1** Two-stage procedure for estimating worst-case subpopulation performance (2)

---

1: INPUT: Subpopulation size $\alpha$, model class $\mathcal{H}$, samples $S_1$ and $S_2$
2: On $S_1$, solve $\widehat{h}_1 \in \operatorname{argmin}_{h \in \mathcal{H}} \frac{1}{|S_1|} \sum_{i \in S_1} (\ell(\theta(X_i); Y_i) - h(Z_i))^2$.
3: On $S_2$, compute the plug-in estimator $\widehat{\mathsf{W}}_\alpha(\widehat{h}_1) = \inf_\eta \left\{ \frac{1}{\alpha|S_2|} \sum_{i \in S_2} \left[ \widehat{h}_1(Z_i) - \eta \right]_+ + \eta \right\}$.

---

**Two-stage procedure** A key remaining challenge in estimating $W_\alpha(\theta)$ is that we can only observe losses $\ell(\theta(X_i); Y_i)$ and never observe the conditional risk $\mu(\cdot)$ (3). We propose a two-stage procedure (Algorithm 1), where we split the sample into two sets $S_1$ and $S_2$. On the first sample $S_1$, we fit an estimator $\widehat{h}_1(Z)$ of the conditional risk $\mu(Z)$, using any model class $\mathcal{H}$ (class of mappings $\mathcal{Z} \to \mathbb{R}$), by solving an empirical approximation to the loss minimization problem

$$\underset{h \in \mathcal{H}}{\text{minimize}} \quad \mathbb{E}\left[(\ell(\theta(X); Y) - h(Z))^2\right]. \tag{6}$$

We denote by $h^\star$ a minimizer of (6); for a sufficiently rich model class $\mathcal{H}$, the minimizer is given by $\mu(Z) = \mathbb{E}[\ell(\theta(X); Y) \mid Z]$. The loss minimization formulation (6) allows the use of any ML estimator, as well as standard tools for model selection (e.g. cross validation). In the second stage, on $S_2$ we construct a plug-in estimator for the dual form (5), under the estimated conditional risk $\widehat{h}_1(\cdot)$. In practice, we switch the roles of $S_1$ and $S_2$ and average the resulting estimates to leverage the entire sample.

To estimate the threshold subpopulation size $\alpha^\star(\theta)$, we simply take the plug-in estimator

$$\widehat{\alpha} := \inf\{\alpha : \widehat{W}_\alpha(\widehat{h}_1) \leq \bar{\ell}\}. \tag{7}$$

Since $\alpha \mapsto \widehat{W}_\alpha(\widehat{h}_1)$ is decreasing, the threshold can be efficiently found by a simple bisection search.

## 3 Convergence guarantees

To *rigorously* verify the robustness of a model prior to deployment, we present convergence guarantees for our estimator (Algorithm 1). In Section 3.1, we first give finite-sample convergence at the rate $O_p(\sqrt{\mathfrak{Comp}_n(\mathcal{H})/n})$, where $\mathfrak{Comp}_n(\mathcal{H})$ is the localized Rademacher complexity [9] of the model class $\mathcal{H}$ for estimating the conditional risk $\mu(Z)$. In some situations, it may be appropriate to define subpopulations ($Z$) over features of an image, or natural language descriptions. For such high-dimensional variables $Z$ and complex model classes $\mathcal{H}$ such as deep networks, the complexity measure $\mathfrak{Comp}_n$ is often prohibitively conservative and renders the resulting concentration guarantee meaningless. In Section 3.2, we provide a finite-sample, data-dependent convergence result that depends only on the out-of-sample error for estimating $\mu(\cdot)$. In particular, the out-of-sample error can grow smaller as $\mathcal{H}$ gets richer, and as a result of hyperparameter tuning and model selection, it is often very small for overparameterized models such as deep networks. This allows us to construct valid finite-sample upper confidence bounds for the worst-case subpopulation performance (2) even when $Z$ is defined over high-dimensional features and $\mathcal{H}$ represent deep networks. Finally, in Section 3.3, we provide convergence guarantees for our estimator (7) for the certificate of robustness (4). By building on previous guarantees, we are again able to obtain both types of results.

We restrict attention to nonnegative and bounded losses.

**Assumption 1.** *There is a $B$ such that $\ell(\theta(X); Y) \in [0, B]$, and $h(Z) \in [0, B]$ a.s. for all $h \in \mathcal{H}$.*

Throughout, we do not stipulate well-specification, meaning that we allow the conditional risk $\mu(\cdot) = \mathbb{E}[\ell(\theta(X); Y) \mid \cdot]$ not to be in the model class $\mathcal{H}$.

### 3.1 Concentration using the localized Rademacher complexity

To characterize the finite-sample convergence behavior of our estimator $\widehat{W}_\alpha(\theta)$, we begin by decomposing the error into two terms relating to the two stages in Algorithm 1. Recalling the notation in Eq. (5) (so that $W_\alpha(\mu) = W_\alpha(\theta)$), we have

$$W_\alpha(\mu) - \widehat{W}_\alpha(\widehat{h}_1) = \underbrace{W_\alpha(\mu) - W_\alpha(\widehat{h}_1)}_{(a):\text{ first stage}} + \underbrace{W_\alpha(\widehat{h}_1) - \widehat{W}_\alpha(\widehat{h}_1)}_{(b):\text{ second stage}}. \tag{8}$$

To bound term $(b)$, we prove concentration guarantees for estimators of the dual (5) (see Proposition 4 in Appendix A.1). To bound term $(a)$, we use a localized notion of the Rademacher complexity.

Formally, for $\xi_1, \ldots, \xi_n \in \Xi$ and i.i.d. random signs $\varepsilon_i \in \{-1, 1\}$ (independent of $\xi_i$), recall the standard notion of (empirical) Rademacher complexity of $\mathcal{G} \subseteq \{g : \Xi \to \mathbb{R}\}$

$$\mathfrak{R}_n(\mathcal{G}) := \mathbb{E}_\varepsilon\left[\sup_{g \in \mathcal{G}} \frac{1}{n} \sum_{i=1}^n \varepsilon_i g(\xi_i)\right].$$

We say that a function $\psi : \mathbb{R}_+ \to \mathbb{R}_+$ is *sub-root* [9] if it is nonnegative, nondecreasing, and $r \mapsto \psi(r)/\sqrt{r}$ is nonincreasing for $r > 0$. Any (non-constant) sub-root function is continuous, and has a unique positive fixed point. Let $\psi_n : \mathbb{R}_+ \to \mathbb{R}_+$ be a sub-root upper bound on the localized Rademacher complexity $\psi_n(r) \geq \mathbb{E}\left[\mathfrak{R}_n \left\{g \in \mathcal{G} : \mathbb{E}[g^2] \leq r\right\}\right]$. (The localized Rademacher complexity itself is sub-root.) The fixed point of $\psi_n$ characterizes generalization guarantees [9, 54].

Let $h^\star$ be the best model in the model class $\mathcal{H}$

$$h^\star := \operatorname*{argmin}_{h \in \mathcal{H}} \mathbb{E}[(\ell(\theta; X, Y) - h(Z))^2].$$

Let $\psi_{|S_1|}(r)$ be a subroot upper bound on the localized Rademacher complexity around $h^\star$

$$\psi_{|S_1|}(r) \geq 2\mathbb{E}\left[\mathfrak{R}_{|S_1|}\left\{h \in \mathcal{H} : \mathbb{E}[(h(Z) - h^\star(Z))^2] \leq rB^2/4\right\}\right]. \tag{9}$$

We define $r^\star_{|S_1|}$ as the fixed point of $\psi_{|S_1|}(r)$.

As we show shortly, we bound the estimation error of our procedure using the *square root* of the excess risk in the first-stage problem (6)

$$\mathbb{E}\left[\left(\ell(\theta; X, Y) - \widehat{h}_1(Z)\right)^2 \mid S_1\right] - \mathbb{E}\left[(\ell(\theta; X, Y) - h^\star(Z))^2\right]$$

By using a refined analysis offered by localized Rademacher complexities, we are able to use a fast rate of convergence of $O_p(\mathfrak{Comp}_n(\mathcal{H})/n)$ on the preceding excess risk. In turn, this provides the following $O_p(\sqrt{\mathfrak{Comp}_n(\mathcal{H})/n})$ bound on the estimation error as we prove in Appendix A.2. In the bound, we have made explicit the approximation error term $\|h^\star - \mu\|_{L^2}$. As the model class $\mathcal{H}$ grows richer, there is tension as the approximation error term will shrink, yet the localized Rademacher complexity of $\mathcal{H}$ will grow.

**Theorem 1.** *Let Assumption 1 hold. For some constant $C > 0$, with probability at least $1 - 2\delta$,*

$$\left|\mathsf{W}_\alpha(\theta) - \widehat{\mathsf{W}}_\alpha(\widehat{h}_1)\right| \leq \frac{CB}{\alpha}\left(\sqrt{r^\star_{|S_1|}} + \sqrt{\frac{\log(1/\delta)}{|S_1|}} + \sqrt{\frac{\log(2/\delta)}{|S_2|}}\right) + \frac{1}{\alpha}\|h^\star - \mu\|_{L_2}.$$

If we let $S_1$ be $(1 - 1/k)$-fraction of the data and $S_2$ be the remaining $1/k$-fraction for some integer $k$ (e.g. $k = 5$), we have $|S_1| \asymp |S_2| \asymp n$. Thus, by controlling the fixed point $r^\star_{|S_1|}$ of the localized Rademacher complexity, we are able to provide convergence of our estimator (3). For example, when $\mathcal{H}$ is a bounded VC-class [81], it is known that its fixed point satisfy [9, Corollary 3.7]

$$r^\star_{|S_1|} \asymp \log(|S_1|/\mathsf{VC}(\mathcal{H})) \cdot \mathsf{VC}(\mathcal{H})/|S_1|,$$

where $\mathsf{VC}(\cdot)$ is the VC-dimension.

## 3.2 Data-dependent dimension-free concentration

In some applications, it may be natural to model $Z$ as a high-dimensional variable. This may include large subsets of $(X, Y)$, or defining $Z$ using unstructured information such as images or natural languages. In these instances, we may wish to use deep networks as the model class $\mathcal{H}$ for estimating the conditional risk (3). We now provide an alternative concentration result that depends on the size of model class $\mathcal{H}$ only through the out-of-sample error in the first-stage problem (6). We denote for simplicity

$$\Delta_S(h) := \frac{1}{|S|}\sum_{i \in S}(\ell(\theta(X_i); Y_i) - h(Z_i))^2. \tag{10}$$

for any function $h : \mathcal{Z} \to \mathbb{R}$ on any data set $S$. We prove the following result in Appendix A.3.

**Theorem 2.** *Let Assumption 1 hold. For some constant $C > 0$, with probability at least $1 - 3\delta$,*

$$|\mathsf{W}_\alpha(\theta) - \widehat{\mathsf{W}}_\alpha(\widehat{h}_1)| \leq \frac{1}{\alpha}\left(\sqrt{\left[\Delta_{S_2}(\widehat{h}_1) - \Delta_{S_2}(h^\star)\right]_+} + CB\left(\frac{\log(1/\delta)}{|S_2|}\right)^{1/4} + \|h^\star - \mu\|_{L^2}\right).$$

*Moreover, if the model class $\mathcal{H}$ is convex, then $\|h^\star - \mu\|_{L^2}$ can be replaced with $\|h^\star - \mu\|_{L^1}$.*

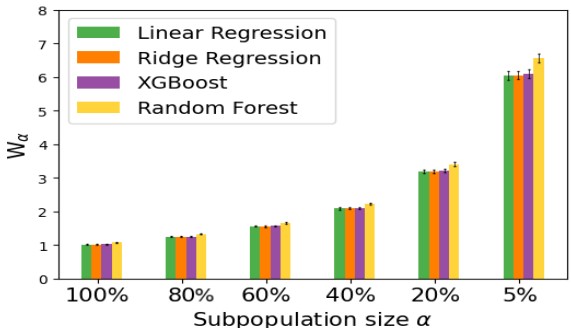

**Figure 2:** Worst-case subpopulation performance $W_\alpha(\theta)$, where $W_{1.0}(\theta) = \mathbb{E}[\ell(\theta(X); Y)]$.

Following convention in learning theory, we refer to our data-dependent concentration guarantee *dimension-free*. For overparameterized model classes $\mathcal{H}$ such as deep networks, the localized Rademacher complexity in Theorem 1 becomes prohibitively large [10, 86]. In contrast, the current result can still provide meaningful finite-sample bounds: model selection and hyperparameter tuning provides low out-of-sample performance in practice, and the difference $\Delta_{S_2}(\widehat{h}_1) - \Delta_{S_2}(h^\star)$ can be often made very small. Concretely, it is possible to calculate an upper bound on this term as $\Delta_{S_2}(h^\star)$ is lower bounded by $\min_{h \in \mathcal{H}} \Delta_{S_2}(h)$.

### 3.3 Certificate of robustness

Instead of estimating the worst-case subpopulation performance for a fixed subpopulation size $\alpha$, it may be natural to posit a level of acceptable performance (upper bound $\bar{\ell}$ on the loss) and study $\alpha^\star(\theta)$, the smallest subpopulation size (4) over which the model $\theta(\cdot)$ can guarantee acceptable performance. Our plug-in estimator $\widehat{\alpha}$ given in Eq. (7) enjoys similar concentration guarantees as those given in Sections 3.1, 3.2. The following theorem—whose proof we give in Appendix A.4—states that the true $\alpha^\star(\theta)$ is either close to our estimator $\widehat{\alpha}$ or it is sufficiently small, certifying the robustness of the model against subpopulation shifts.

**Theorem 3.** *Let Assumption 1 hold, let $U(\delta) > 0$ be such that for any fixed $\alpha \in (0, 1]$, $|\widehat{W}_\alpha(\widehat{h}) - W_\alpha(\theta)| \leq U(\delta)/\alpha$ with probability at least $1 - \delta$. Then given any $\underline{\alpha} \in (0, 1]$, either $\alpha^\star(\theta) < \underline{\alpha}$, or*

$$\left| \frac{\alpha^\star(\theta)}{\widehat{\alpha}} - 1 \right| \leq \frac{U(\delta)}{\widehat{\mathbb{E}} \left[ \widehat{h}(Z) - \widehat{P}^{-1}_{1 - \underline{\alpha} \wedge \widehat{\alpha}} \left( \widehat{h}(Z) \right) \right]_+}$$

*with probability at least $1 - \delta$, where $\widehat{\mathbb{E}}$ and $\widehat{P}^{-1}_{1-\alpha}$ denote the expectation and the $(1 - \alpha)$-quantile under the empirical probability measure induced by $S_2$.*

Our approach simultaneously provides localized Rademacher complexity bounds and dimension-free guarantees. Our bound becomes large as $\underline{\alpha} \to 0$ and we conjecture this to be a fundamental difficulty as the worst-case subpopulation performance (2) focuses on $\alpha$-faction of the data.

## 4 Experiments

On two real datasets, we demonstrate that our diagnostic allows certifying model performance across subpopulations. We first study a drug dosage prediction problem, where our procedure ascertains the robustness of a linear regression model over substantially more expressive model classes. Then, we turn to a large-scale computer vision application based on satellite images [25] where natural distribution shifts were recently studied [53]. In both settings, we illustrate how our worst-case subpopulation approach raises awareness on brittle models without knowledge of out-of-distribution samples. Finally, to verify asymptotic convergence of our proposed two-stage estimator, we present a simultion experiment on a classification task in Appendix C. For all experiments, we use gradient boosted decision trees (package XGBoost [22]) to estimate the conditional risk $\mu(Z) = \mathbb{E}[\ell(\theta(X); Y) \mid Z]$.

### 4.1 Warfarin

Warfarin is one of the most widely used anticoagulant, often prescribed to prevent strokes [26]. Its optimal dosage varies substantially across genetics, demographics, and existing conditions (up to ten times). We study a Pharmacogenetics and Pharmacogenomics Knowledge Base dataset constructed from optimal dosages found through trial and error by clinicians. The dataset comprises of 4,788 patients (after excluding missing data) alongside features representing demographics, genetic markers, medication history, pre-existing conditions, and reason for treatment. Consortium [26] found that a linear model outperforms a number of more complicated modeling approaches (e.g. kernel methods, neural networks, splines, boosting) for predicting the optimal dosage.

Such average-case performance needs to translate uniformly to different subpopulations; we need to ensure automated medical models perform well on underrepresented groups [20, 63, 40, 2]. We wish to evaluate and compare the worst-case subpopulation performance of different models over $Z = X$, the entire feature vector. Following Consortium [26], we take the root-dosage as our outcome $Y$, and consider the squared loss $\ell(\theta(X); Y) = (Y - \theta(X))^2$. In Figure 2, we observe that the linear model closely matches the performance of more expressive models even over small subpopulations. Moreover, the trend holds over a range of different subpopulation sizes (up to $\alpha = 5\%$). Our finding instills confidence in the linear regression model: in addition to being simple and interpretable, our diagnostic certifies its advantageous performance even on tail subpopulations. However, our diagnostic raises some concerns about poor subpopulation performance: on $\alpha = 5\%$ of the training population, all models suffer prediction error six times worse than the average-case performance.

### 4.2 Functional Map of the World (FMoW)

Satellite images can impact economic and environmental policies globally by allowing large-scale measurements on poverty [1], population changes, deforestation, and economic growth [42]. An automated approach allows analyzing data from remote regions at a relatively low cost and provides continuous monitoring of land usage. Towards this goal, it is critical that the models perform reliably across time and space. We study this problem on the Functional Map of the World (FMoW) dataset [25], where the goal is to predict building / land use categories (62 classes) based on satellite images. Across different models, we observe that performance remains similar either temporally or spatially when each dimension is considered *separately*, but there is substantial variability across intersections of region and year. For a standard *DenseNet ERM* model [47, 53] that achieves near-state-of-the-art performance, we present these trends in Figure 3(a). In Figure 3(b), we observe substantial variability in classwise error rates; there is a varying level of difficulty across different classes. (We observed similar patterns for other models.)

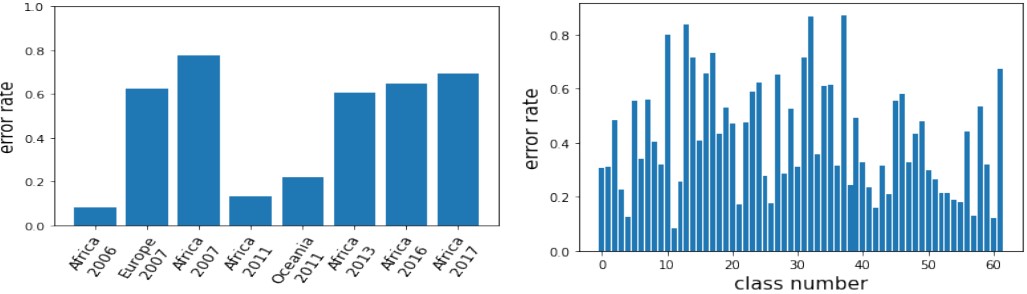

**Figure 3:** For *DenseNet ERM*, spatiotemproal intersectionality (left) and performance by class (right)

We take the perspective of an analyst evaluating prediction models for land usage, based on data collected during 2002-2013. The FMoW dataset provides fertile grounds for demonstrating our method as it includes natural distribution shifts [53], both spatial and temporal. In particular, we demonstrate model robustness on out-of-distribution samples collected in 2016-2018. On validation data collected during 2002-2013, we first evaluate model performance on subpopulations defined across metadata on a satellite image, which consists of (subsets of) {*longitude, latitude, cloud cover, region, year*} and the label $Y$. Then, we observe how our procedure selects models that perform well "in the future" without requiring out-of-distribution data.

We examine a range of different models trained on the FMoW-WILDS training set (collected in 2002-2013, $n$ =76,863) which fall into two broad categories. First, we consider models pre-trained on ImageNet and finetuned on the FMoW training set. These include *DenseNet* models trained using ERM and the recently proposed invariant risk minimization (IRM) framework [6]. We also study the Dual Path Network-68 (*DPN-68*) model with connection paths that enable feature reusage and feature exploration proposed by Chen et al. [23]. We use *DPN-68* trained on FMoW using ERM as reported in [58]. These models all achieve in distribution (ID) accuracy of $\sim 60\%$ on a heldout validation set ("ID val", collected in 2002-2013, $n$ =11,483).

Second, we consider models derived from the recently proposed CLIP model [62], which was trained on large and heterogeneous data sources comprising of 40M image-text pairs using natural language supervision and contrastive losses. The pre-training data for CLIP is 400 times bigger than ImageNet, and Radford et al. [62] have observed that zero-shot applications of CLIP exhibits substantial *relative robustness gains* over other state-of-the-art methods on natural distribution shifts of ImageNet.

However, on the FMoW in-distribution (2002–2013) validation set, zero-shot CLIP only achieves 19.3% accuracy compared to the 60% accuracy of ImageNet pre-trained models. We thus finetune it using satellite images in the FMoW training data. While finetuning substantially improves ID accuracy on FMoW to 70.2%, the relative robustness gains of the zero-shot CLIP model severely degrade. To address this problem, Wortsman et al. [85] proposed a weight-space ensembling method (*CLIP WiSE-FT*) where they average the network weights of the zero-shot CLIP model and its finetuned counterpart. These ensembled networks have been observed to exhibit large Pareto improvements in both in-distribution and out-of-distribution accuracy, including on the FMoW dataset.

Motivated by the observed robustness gains, we average the network weights $\theta_0$ of the *CLIP Zeroshot* model and that of *CLIP fine-tuned* $\theta_1$ to generate a new network $(1 - \lambda)\theta_0 + \lambda\theta_1$, where $\lambda \in [0, 1]$ controls how much weight is given to the task-specific, fine-tuned model (domain expertise). We select $\lambda = 0.4$ so that the ensembled model (*CLIP WiSE-FT*) achieves similar performance as ImageNet pre-trained counterparts on the in-distribution validation data. To further make models comparable with respect to the cross entropy loss, we calibrate the *CLIP WiSE-FT* model by tuning the temperature parameter so that its average loss on the in-distribution validation set matches the worst average loss of ImageNet pre-trained models (*DenseNet ERM*). See Appendix B for detailed experimental settings and training specifications.

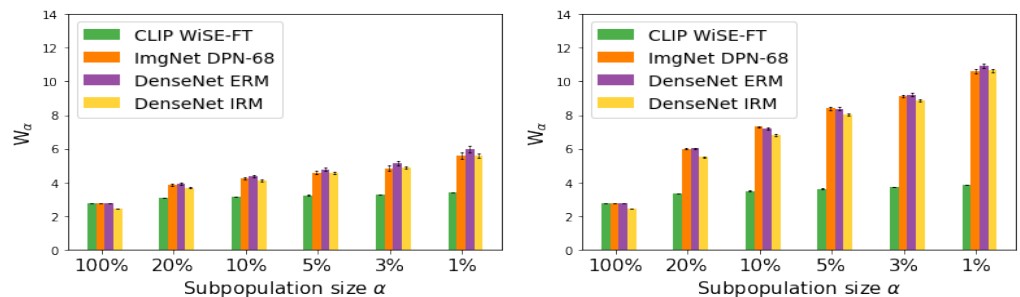

**Figure 4.** Left: $Z$ = (all metadata); Right: $Z$ = (all metadata, Y). Results are averaged over 50 random seeds with error bars corresponding to a 95% confidence interval over the random runs.

We compute estimators of $W_\alpha$ (Algorithm 1) on the in-distribution validation data (ID val) using the standard cross entropy loss. In Figure 4, we summarize the estimated worst-case subpopulation performances defined over the entire *metadata*, across different subpopulation sizes $\alpha$. First, we note that all models have comparable in-distribution accuracy of $\sim 60\%$ and *DenseNet IRM* has the best average-case cross entropy loss. However, the worst-case subpopulation performance of the ImageNet pre-trained models is substantially worse compared to that of *CLIP WiSE-FT*. This gap grows larger as the subpopulation size $\alpha$ becomes increasingly small. Evaluations on worst-case subpopulations suggest that *CLIP WiSE-FT* exhibits robustness against subpopulation shifts; in contrast, average-case evaluations will select *DenseNet IRM*.

We observe a drastic performance deterioration on tail subpopulations. The inclusion of label information in $Z$ significantly deteriorates worst-case performance, raising concerns about the distributional robustness of all models including changes in the label distribution. In Table 1, we present model performances on the out-of-distribution ("future") data collected during 2016–2018.

| Models | ID, 2002–2013 | | OOD, 2016–2018 | | | |
| | Accuracy | Loss | Accuracy | Loss | Africa Accuracy | Africa Loss |
|---|---|---|---|---|---|---|
| CLIP WiSE-FT | 0.61 | 2.78 | 0.56 | 2.84 | 0.38 | 3.08 |
| DenseNet ERM | 0.61 | 2.78 | 0.53 | 3.50 | 0.33 | 5.41 |
| DenseNet IRM | 0.59 | 2.44 | 0.51 | 2.94 | 0.31 | 4.46 |
| DPN-68 | 0.61 | 2.75 | 0.53 | 3.55 | 0.31 | 5.61 |

**Table 1.** Model performance on ID val and OOD test sets. All models suffer a performance drop on the OOD test set in both accuracy and average loss. The performance degradation is particularly significant on images from Africa. On the OOD data, *CLIP WiSE-FT* outperforms other models both in average accuracy/loss and worst-region accuracy/loss.

All models suffer a significant performance drop under temporal distribution shift, particularly on images collected in Africa where predictive accuracy drops by up to 20 percentage points. *CLIP WiSE-FT* exhibits the most robustness under spatiotemporal shift than any other model, as presaged by evaluations of worst-case subpopulation performance in Figure 4.

A key advantage of our method is the flexibility in the choice of $Z$; the modeler can define granular or coarse subpopulations based on this choice. As defining subpopulations over all metadata can be conservative, we present additional results under $Z =$(*region*, *year*) and $Z =$(*region*, *year*, *label* $Y$) in Appendix B. Instead of incorporating labels as a category, it may be more informative to use the *semantic meaning of each class label*. We generate natural language description of the labels by concatenating each class label with engineered prompts, and pass it to the CLIP text encoder [62] to generate a feature representation for the label. In Appendix B.3, we present evaluation results where we take the feature vector in place of the label $Y$ when defining $Z$.

## 5   Discussion

To ensure models perform reliably under operation, we need to *rigorously* certify their performance under distribution shift prior to deployment. We study the *worst-case subpopulation performance* of a model, a natural notion of model robustness that is easy to communicate with users, regulators, and business leaders. Our approach allows flexible modeling of subpopulations over an arbitrary variable $Z$ and automatically accounts for complex intersectionality. We develop scalable estimation procedures for the worst-case subpopulation performance (2) and the certificate of robustness (4) of a model. Our convergence guarantees apply even when we use high-dimensional inputs (e.g. natural language) to define $Z$. Our diagnostic may further inform data collection and model improvement by suggesting data collection efforts and model fixes on regions of $\mathcal{Z}$ with high conditional risk (3).

The worst-case performance (2) over mixture components as subpopulations (1) provides a strong guarantee over arbitrary subpopulations, but it may be overly conservative in cases when there is a natural geometry in $Z \in \mathcal{Z}$. Incorporating such problem-specific structures in defining a tailored notion of subpopulation is a promising research direction towards operationalizing the concepts put forth in this work. As an example, Srivastava et al. [76] recently studied similar notions of worst-case performance defined over human annotations.

We focus on the narrow question of evaluating model robustness under distribution shift; our evaluation perspective is thus inherently limited. Data collection systems inherit socioeconomic inequities, and reinforce existing political power structures. This affects *all* aspects of the ML development pipeline, and our diagnostic is no panacea. A notable limitation of our approach is that we do not explicitly consider the power differential that often exists between those who deploy the prediction system and those for whom it gets used on. Systems must be deployed with considered analysis of its adverse impacts, and we advocate for a holistic approach towards addressing its varied implications.

