# OpenReview forum: "Evaluating model performance under worst-case subpopulations"
_NeurIPS.cc/2021/Conference — NeurIPS 2021 Poster_

### Official Review · Reviewer_jVDR · 2021-07-16

**Rating:** 7
**Confidence:** 4

**Summary:**

The authors developed the evaluation method for the worst-case loss of a given predictor over all the subpopulations. The method is motivated to evaluate how a given predictor is robust against population change of the random variable Z.  The authors theoretically investigate the estimation error of the proposed method and show two high probability bounds on the estimation error. One of the bound depends on the complexity of the model to estimate the conditional mean of loss; however, we can achieve a convergence rate faster than $1/n^{1/4}$ if the model complexity is low. Another one is the data-dependent bound and is independent of the model complexity; however, the convergence rate is $1/n^{1/4}$. They also provide some showcases to utilize the proposed methods.

**Limitations And Societal Impact:**

The limitations and potential societal impact are adequately discussed.

**Main Review:**

This paper is well-written, technically sound, and interesting. The data-dependent bound in Theorem 2 is very surprising for me because it is independent of the model complexity and is only dependent on the learned model. This is quite nice as we can employ an arbitrary complex model. Therefore, my current decision is acceptance.

The paper may be improved by providing a more detailed comparison between the obtained bounds, i.e., those in Theorem 1 and Theorem 2. Specifically, it is better to clarify when the sample complexity obtained from Theorem 2 is better than that obtained from Theorem 1. The authors already mentioned that Theorem 2 is better when the model is too complex. However, it is better to clarify when the model is too complex to make the bound in Theorem 1 loose. It helps readers to understand why the authors provide two bounds.

I checked all the proofs provided in the supplementary material and only found a minor flaw that may be fixed easily. See the following for the detail.
- The right-hand side of Eq. 17 in the appendix might be incorrect but can be fixed easily. The first term should be multiplied by 2, and in the second term, \hat{h} and h^* should be exchanged.


**Time Spent Reviewing:**

7

---

> ### Author Response · Authors · 2021-08-10
> **Response to Reviewer jVDR**
>
> Thank you for your thoughtful and positive feedback. It is much appreciated. Please refer to the above central response for a discussion of Theorem 2, including our explanation on its proof.
>
> We will make sure to better contrast the differences between Theorems 1 and 2. For deep networks, typical generalization bounds such as Theorem 1 that are based on the Rademacher complexity become vacuous due to the high-dimensional nature [1, 2]. In these scenarios, Theorem 2 provides an attractive *dimension-free* alternative, as the out-of-sample error $\Delta_n$ is typically small for rich model classes such as neural networks. Consider, for example, ImageNet-scale settings where we wish to guarantee robustness over distribution shifts in the image distribution. Models typically have hundreds of millions of parameters (e.g. ResNet, vision transformers) but the dataset size is in the order of 1-2 million. Our Theorem 1 will provide vacuous generalization guarantees in such high-dimensional scenarios, yet Theorem 2 will still give useful bounds as modern architectures achieve small out-of-sample error $\Delta_n$.
>
> ### References
>
> [1] C. Zhang, S. Bengio, M. Hardt, B. Recht, and O. Vinyals. Understanding deep learning requires
> rethinking generalization. In Proceedings of the Fifth International Conference on Learning
> Representations, 2017.
>
> [2] P. Bartlett, D. J. Foster, and M. Telgarsky. "Spectrally-normalized margin bounds for neural networks." arXiv preprint arXiv:1706.08498, 2017.

---

> > ### Comment · Reviewer_jVDR · 2021-09-02
> > **Thank you for your response**
> >
> > Thank you for the response and the clarifications.

---

### Official Review · Reviewer_KadU · 2021-07-16

**Rating:** 6
**Confidence:** 3

**Summary:**

The paper provides a method for evaluating the performance of a learned model on worst-case subpopulations of the data. The method is highly scalable and can handle continuous variables.

**Limitations And Societal Impact:**

Nothing comes to mind.

**Main Review:**

$\textbf{Originality}$

To my awareness the presented method is novel. In addition, as pointed out by the authors, the method is highly scalable, unlike previous distributionally robust algorithms.

$\textbf{Quality}$

The work appears technically sounds and proofs of all results are presented in the supplementary material.

$\textbf{Clarity}$

Overall, the paper is well-written, however my main concerns are related to the lack of explicit data generation model and to the notion of worst-case subpopulation risk. In particular, in the introduction the objective (2) is presented, but several important aspects remain unclear to me, also throughout the paper. In particular, what is the distribution $P_Z$ and how does it relate to the data generating distribution? Is it the marginal distribution of $Z$? In this case, how is the distribution $(X,Y)|Z$ defined when the distribution of $Z$ changes?

Additionally, while the notion of subpopulation used is relatively natural, I wonder if equation (1) can be accompanied by some example (e.g. specific application with specified $X,Y,Z, Q_Z$ etc.), to improve readability.

$\textbf{Significance}$

Evaluating model performance under worst-case subpopulations is certainly important for the sake of both robustness and fairness of ML algorithms. In its current form, the method only addresses the problem of estimating whether a given model has a good subpopulation performance. It will be interesting to know if the method can also be used iteratively at training time to help learning a distributionally robust model.


**Time Spent Reviewing:**

3

---

> ### Author Response · Authors · 2021-08-10
> **Response to Reviewer KadU**
>
> Thank you for your thoughtful comments. Please refer to the above central response for why training with respect to the worst-case subpopulation performance cannot scale both computationally and statistically. In it, we also outline our primary motivation for focusing on the narrower question of model evaluation and how this allows us to develop methods for evaluating even the largest computer vision models. We will make sure to explicitly discuss this early on in the introduction, and clearly demarcate the focus of this paper: scalable evaluation of worst-case subpopulation performance.
>
> ### Clarifying the worst-case region
> We appreciate your suggestions to clarify the notion of subpopulations that we formulate. We will include more elaborated discussions with accompanying examples. First, we note that the (data-generating) training distribution $P$ can be decomposed as $P = P_{(X, Y) | Z} \times P_Z$. In our definition of worst-case subpopulation performance, we assume that the conditional distribution $P_{(X, Y) | Z}$ is fixed, and consider distribution shifts in the marginal distribution $P_Z$. In particular, we study subpopulations of the training distribution $P_Z$, formally defined as mixture components of the probability measure. As it is unrealistic to consider arbitrarily small subpopulations&mdash;there will be no training data for such a subpopulation&mdash;we consider mixture components with mixture weight larger than a user-specified proportion $\alpha$. Please note that we do not impose any structural assumptions on the data-generation model&mdash;such an assumption would be prohibitively restrictive. Furthermore, our above approach is without loss of generality, as $Z$ can be chosen arbitrarily by the analyst.
>
> As suggested, we will illustrate our notion of subpopulations with a concrete example. Suppose a simple setting where we wish to predict income ($Y$) based on race and age ($X$). We wish to provide good performance over age ($Z$), where the training data for $Z$ is distributed as $Z \sim N(30, 10^2) = P_Z$. We consider any subpopulation of the training population in the age space, meaning all PDFs $q_Z(\cdot)$ such that $q_Z(z) / p_Z(z) <= 1/\alpha$, where $p_Z(\cdot)$ is the PDF of the normal distribution $N(30, 10^2)$. We will illustrate such examples with appropriate graphics in the final version (unfortunately, we cannot attach our visualizations on OpenReview).

---

> > ### Comment · Reviewer_KadU · 2021-08-26
> > **Thank you for your response!**
> >
> > Dear authors,
> >
> > Thank you for the response and the clarifications. The provided example and clarifications are indeed very helpful and I'm looking forward to seeing them integrated in a next version of the paper.

---

### Official Review · Reviewer_o544 · 2021-07-16

**Rating:** 6
**Confidence:** 2

**Summary:**


This paper poses the problem of evaluating the worst-case risk over subpopulations, which is an important way to quantify the fairness of a learning algorithm. Unfortunately they do not consider the learning aspect of this problem, but rather treat the model as fixed, however they do make substantial contributions, and show that even evaluating worst-case subpopulation risk is challenging in full generality.  It seems the future work could address this shortcoming. In particular, in their model, they allow for a sophisticated and abstract notion of subpopulations, defined by a function family $\cal H$ applied to a random variable $Z$, usually representing protected information. The authors explore an approach based on localized Rademacher averages, and one using a 2-fold cross-validation strategy, to control for the multiple comparisons aspect of considering many subpopulations in the worst-case.


**Limitations And Societal Impact:**

Yes

**Main Review:**




I found the problem to be appealing, though I would have preferred to see some analysis of the learning aspects of the question, since as it stands only the statistical estimation aspect of the problem is really handled. In particular, as the authors already consider bisecting the sample to evaluate worst-case population risk, it stands to reason that one could trisect it to train a model on the first fold, and use the next two folds to evaluate worst-case risk, though it would be more satisfying if all of these tasks could be accomplished on a single sample.  Unfortunately I found the paper rather difficult to follow for several reasons, not the least of which was the lingering persistent surprise that the learned function was fixed, and the hypothesis class is only over subpopulations. In particular, further exposition of the hypothesis class $\cal H$, and how it can be used to model various subpopulation categories, would have been greatly appreciated.


I also wonder whether the localized Rademacher average methods were only loose due to poor constants in the cited analyses; I believe [2] substantially improves the earlier results of Bartlett, Koltchinskii, and others. Furthermore, the decision to use localized Rademacher averages, rather than, e.g., global Rademacher averages was not well-explained.  I suspect this decision was made quite consciously, as the authors seem very knowledgeable in statistical learning theory, but to a layreader, it could easily come across as somewhat arbitrary (and thus somewhat uninteresting).  To clarify, in my understanding, localized bounds are very appropriate for this purpose, as they are more sensitive to low-frequency (low raw-variance) events, and behave essentially like relative bounds, as opposed to uniform additive bounds. Some comparison to methods using global Rademacher averages and an explanation of how this is relevant to fairness w.r.t. small subpopulations, would help connect this section better to the overall themes of the paper, and to ideas in statistical learning.

I believe a minor technical error is to be found below, though it does not significantly impact the work (I don’t believe localized Rademacher averages were used in the experiments anyway).  On line 201: “The localized Rademacher complexity itself is sub-root.” Doesn’t the function family need to be closed under absolute convex combination (a.k.a., take the star convex hull with 0), for the localized Rademacher complexity to be sub-root? I'm reasonably confident that this is the case, otherwise localized Rademacher averages are 0 or undefined for sufficiently small $r$, as the class may become empty.


I would also have liked to see some experiments with simpler hypothesis classes (like threshold families), using the localized Rademacher average bounds as well. Even if they were vacuous or not very sharp, it would have been interesting, and they could have been compared to the cross-validation results. Simpler hypothesis classes would have also nicely bridged this work to earlier results based on, e.g., demographic groups, as per the example of figure 1.




Specifics:

A comparison to [1] would be quite interesting. In the parlance of that work, it seems the authors wish to compute the egalitarian malfare, and their proof techniques revolves around subpopulation Rademacher averages, which may be relevant.  They also consider other power-means, which may be possible in this framework as well.

Line 68: what are the implications of this averaging with respect to various loss functions? This feels a lot like 2-fold cross-validation.

86: all subpopulations large than -> all subpopulations larger than


131/133: Klein (not Consortium), is the lead author of [27].

156: I followed $h^*(Z)$ and $\hat{h}(Z)$, but it took quite some time to understand $h(Z)$. It's really not clear what the model class here is doing. Is it just that $\cal H$ needs to be able to predict conditional risk of values given $Z$? This is in dire need of more exposition and explanation.

Equation 6: This was quite difficult to follow how this translates to worst-case subpopulation risk.  More explanation would be appreciated as this is a crucial detail.

Equation 6: Of note is that if we minimize over $\theta$ instead of $\cal{h}$, it seems we are trying to minimize the weighted average of conditional variances over losses, conditioned on each $Z$? This seems quite related to maximizing within-group fairness (while completely neglecting between group fairness).  Is there an interesting connection to be made here?

It's clear to me that the objective (6) is a standard square loss minimisation problem, but what I don't understand is how large is the error due to using the plug-in estimator (lemma 1), and in particular, in the comment on 167, “for a sufficiently rich model class $\cal H$”, what happens if $\cal H$ is not sufficiently rich.


Overall, why are we not simply maximizing conditional risk values given $Z$? I get that that becomes statistically intractable for a large $Z$, but if $Z$ is too large, then presumably we don't have a sufficiently rich $\cal H$ class, so the relationship between this minimizer and worst-case subpopulation risk breaks down anyway. What am I missing here, how do we overcome that issue?

Going into algorithm 1 on my first read, I really thought we were talking about minimization over the model. Some more explanation of how the class of $\cal H$ is a class over subpopulations, and we're just trying to estimate the worst case risk for a fixed model would be very helpful.


167: “the minimizer is given by” Isn’t this a scalar? Is it the minimum or minimizer?

Algorithm 1 line 3: without some parentheses, the $\eta$ may be read as though it is inside the sum, which is of course foolish, and I believe inconsistent with equation 5.  This should be written more clearly.



229: This is a bit difficult to read.

Section 3.2: I struggle to see how we can get any kind of guarantee using a large deep network for $\cal H$, unless we're assuming we can train the network to optimality. It looks to me like the gap is measured between the selected $\hat{h}$ and the worst-case subpopulation, but it feels like it should be that the estimate on S1 of $\hat{h}$ should roughly be an upper bound, and the estimate on S2 of $\hat{h}$ should roughly be a lower bound on the true worst-case subpopulation risk.  I don’t really see why localization assumptions 1, 2 are needed in theorem 2.


References:


[1] An Axiomatic Theory of Provably-Fair Welfare-Centric Machine Learning
C Cousins

[2] Local rademacher complexity: Sharper risk bounds with and without unlabeled samples
L Oneto, A Ghio, S Ridella, D Anguita

**Time Spent Reviewing:**

8

---

> ### Author Response · Authors · 2021-08-10
> **Response to Reviewer o544**
>
> Thank you for your detailed review. Please refer to the above central response for why training with respect to the worst-case subpopulation performance cannot scale both computationally and statistically. In it, we also outline our primary motivation for focusing on the narrower question of model evaluation and how this allows us to develop methods for evaluating even the largest computer vision models. We will make sure to explicitly discuss this early on in the introduction, and clearly demarcate the focus of this paper: scalable evaluation of worst-case subpopulation performance.
>
> ### Local vs global Rademacher complexity
> We specifically chose the localized Rademacher complexity method by Bartlett et al. [1], as it provides substantially faster rates compared to the global Rademacher analysis. By focusing a small subset of the model class, we are able to show a fast rate of convergence for estimating $\mathbb{E}[\ell(\theta; X, Y) | Z]$ (using squared error) at the rate $O_p(1/n)$, instead of the usual slow rate of convergence $O_p(n^{-½})$ provided by global Rademacher complexities [2]. As the square root of this quantity links to estimation error for the worst-case subpopulation population, this allows us to get the final $O_p(n^{-½})$ rate in Theorem 1. Had we used global Rademacher complexities, this final rate would deteriorate to $O_p(n^{¼})$. We will discuss this context more explicitly in the camera-ready version.
>
> ### Request for clarification
> We were hoping if the reviewer could clarify what they meant by the below comment.
>
> > I would also have liked to see some experiments with simpler hypothesis classes (like threshold families), using the localized Rademacher average bounds as well. Even if they were vacuous or not very sharp, it would have been interesting, and they could have been compared to the cross-validation results. Simpler hypothesis classes would have also nicely bridged this work to earlier results based on, e.g., demographic groups, as per the example of figure 1.
>
> In particular, it is unclear to us what it means to use “threshold families” as a hypothesis class. We consider all subpopulations defined over $Z$, which can include demographic groups defined over a threshold rule (e.g., $Z_1 > 4$) or any combination thereof.
>
> ### Response to specific comments
>
> Thank you for the detailed comments. We will incorporate suggestions without mention, and clarify or discuss important points below.
>
> > A comparison to [1] would be quite interesting. In the parlance of that work, it seems the authors wish to compute the egalitarian malfare, and their proof techniques revolves around subpopulation Rademacher averages, which may be relevant. They also consider other power-means, which may be possible in this framework as well.
>
> Thank you for bringing this work to our attention&mdash;we will include it in our discussion of related literature. The main difference between this work and our approach is that the worst-case subpopulation performance considers an infinite number of subpopulations and does not rely on pre-defined groups. This allows us to automatically consider potentially complicated intersectionality between features. The welfare maximization approach, in contrast, considers finitely many pre-defined subgroups.
>
> > Line 68: what are the implications of this averaging with respect to various loss functions? This feels a lot like 2-fold cross-validation.
>
> We are indeed using the idea of 2-fold CV to fully utilize the data. We will make this connection more explicit, and present a general K-fold CV procedure in lieu of Algorithm 1. Our procedure has intellectual connections with the recently proposed cross-fitting procedure [3] for semiparametric inference.
>
> > 156: I followed $h^∗(Z)$ and $h^(Z)$, but it took quite some time to understand $h(Z)$. It's really not clear what the model class here is doing. Is it just that $\mathcal H$ needs to be able to predict conditional risk of values given $Z$? This is in dire need of more exposition and explanation.
>
> Yes, $h(Z)$ is a dummy predictor for the conditional risk $h^*(Z) = \mathbb{E}[\ell(\theta; X, Y) | Z]$, where we denote the prediction model class as $\mathcal{H}$.
>
> > Equation 6: Of note is that if we minimize over $\theta$ instead of $h$, it seems we are trying to minimize the weighted average of conditional variances over losses, conditioned on each $Z$? This seems quite related to maximizing within-group fairness (while completely neglecting between group fairness). Is there an interesting connection to be made here?
>
> If you are referring to the connection between distributionally robust optimization and variance regularization (e.g., Duchi and Namkoong [4]), this equivalence only holds when $\alpha$ is very close to 100%. That is, when the worst-case subpopulation performance is close to the average-case performance. This connection to within-group variance, however, does not hold in most application scenarios of interest as we are typically interested in values of $\alpha$ far from 100%.
>
> > It's clear to me that the objective (6) is a standard square loss minimisation problem, but what I don't understand is how large is the error due to using the plug-in estimator (lemma 1), and in particular, in the comment on 167, “for a sufficiently rich model class $\mathcal H$”, what happens if $\mathcal H$ is not sufficiently rich.
>
> As the reviewer notes, the misspecification error will be a problem. In practice, procedures such as cross-validation allow us to choose sufficiently rich model classes $\mathcal{H}$. We will update both Theorems 1 and 2 to better characterize the dependence on the misspecification error.
>
> When there is model misspecification (i.e., if Assumption 2 doesn’t hold), we can still prove a variant of Theorem 1. We will update our statement of Theorem 1 to incorporate this flexibility. We have the same bound as before, but with an additional term representing misspecification error. More formally, if we denote the best possible estimator in the model class by $\tilde{h} := \arg\min_{h\in\mathcal{H}} \mathbb{E} (\ell(\theta(X);Y) - h(Z))^2$, then the RHS of Lemma 8 would have an extra term of $\mathbb{E}(\tilde{h} - h^\star)^2$, which measures the misspecification error of the model class and, in particular, equals zero under well-specification. The local Rademacher complexity we used for $r^\star$ would be centered around $\tilde{h}$ instead. All other parts of the proof continue to hold.
>
> On the other hand, Theorem 2 does not rely on Assumption 2 at all as noted in our central response, and uses the out-of-sample error $\Delta_n$ to characterize the estimation error.
>
> > Overall, why are we not simply maximizing conditional risk values given $Z$? I get that that becomes statistically intractable for a large $Z$, but if $Z$ is too large, then presumably we don't have a sufficiently $\mathcal H$, class anyway so the relationship between this minimizer and worst-case subpopulation risk breaks down anyway. What am I missing here, how do we overcome that issue?
>
> Maximizing the conditional risk $\mathbb{E}[\ell(\theta; X, Y)|Z]$ is a high-dimensional non-concave maximization problem. We do not expect to be able to provide any meaningful guarantee over this problem. On the other hand, the worst-case optimization problem (2) over probability measures is a concave maximization problem. By using convex duality, we are able to provide a tractable procedure in Section 2 of our paper. We believe this is crucial: models may be trained heuristically, but they need to be evaluated in a rigorous and principled manner.
>
> Note also that the difference between optimizing over $Z$ and over distributions of $Z$ is the difference between robust optimization and distributionally robust optimization. The DRO framework turns an intractable finite-dimensional non-concave maximization problem to an infinite-dimensional concave maximization problem, which can often be solved efficiently using dual methods.
>
> > Section 3.2: I struggle to see how we can get any kind of guarantee using a large deep network for $\mathcal H$, unless we're assuming we can train the network to optimality.
>
> Please see our discussion of Theorem 2 in the central response. For deep networks, typical generalization bounds such as Theorem 1 based on the Rademacher complexity become vacuous due to the high-dimensional nature. In these scenarios, Theorem 2 provides an attractive *dimension-free* alternative, as the out-of-sample error $\Delta_n$ is typically small for rich model classes such as neural networks. Consider, for example, ImageNet-scale settings where we wish to guarantee robustness over distribution shifts in the image distribution. Models typically have hundreds of millions of parameters (e.g. ResNet, vision transformers) but the dataset size is in the order of 1-2 million. Our Theorem 1 will provide vacuous generalization guarantees in such high-dimensional scenarios, yet Theorem 2 will still give useful bounds as modern architectures achieve small out-of-sample error $\Delta_n$.
>
> ### References
> [1] P. L. Bartlett, O. Bousquet, and S. Mendelson. "Local Rademacher Complexities." The Annals of Statistics 33.4 (2005): 1497-1537.
>
> [2] M. J. Wainwright. High-dimensional statistics: A non-asymptotic viewpoint. Vol. 48. Cambridge University Press, 2019.
>
> [3] Victor Chernozhukov, Denis Chetverikov, Mert Demirer, Esther Duflo, Christian Hansen, Whitney Newey, James Robins, Double/debiased machine learning for treatment and structural parameters, The Econometrics Journal, Volume 21, Issue 1, 1 February 2018, Pages C1–C68, https://doi.org/10.1111/ectj.12097
>
> [4] J. Duchi and H. Namkoong. Variance-based Regularization with Convex Objectives. Journal of Machine Learning Research, 20(68):1-55, 2019.

---

> > ### Comment · Reviewer_o544 · 2021-08-13
> > **Threshold Functions**
> >
> > Apologies, by threshold families I meant *univariate decision stumps* or *access aligned hyperplanes*. These are contained in the set of all subpopulations, but it is (usually) a much smaller family, so I would expect less overfitting, and thus the localized Rademacher bounds may perform substantially better.

---

> > > ### Author Response · Authors · 2021-08-15
> > > **Follow-up response to Reviewer o544**
> > >
> > > Thank you for the clarification. This is indeed an interesting suggestion; we fully agree with the high-level goal of the reviewer's comment.
> > >
> > > We would like to first note that for any fixed prediction model $\theta$, it is often unclear what the functional form of the conditional risk $\mathbb{E}[\ell(\theta(X); Y) | Z]$ will look like. For example, it may be unrealistic to assume that the conditional risk can be approximated by univariate decision stumps over Z. However, it may often be realistic to posit that the conditional risk belongs to a reproducing kernel Hilbert space (RKHS), a flexible space of *nonparametric* functions. Lemma 2 provides informative bounds on the localized Rademacher complexity for this flexible model class. In particular, our result implies that for kernels with a exponentially decaying eigenvalues (e.g. Gaussian kernel), the localized Rademacher complexity scales as $B^2 \sqrt{\log(n)}/n$ where n is the number of samples, and $B$ is the maximal RKHS norm of the model class. Our analysis applies to a much broader class of kernels, and builds on the approach of Mendelson (2003). We will make sure to expand our discussion for the overall context for this result in the final version of the paper.
> > >
> > > Again, we appreciate the thoughtful suggestions. Please refer to our centralized response as well as the tailored reply above and let us know if you have any other questions/concerns/suggestions.
> > >
> > > Mendelson, Shahar. On the performance of kernel classes. *Journal of Machine Learning Research*. 2003.

---

### Official Review · Reviewer_3a1D · 2021-07-18

**Rating:** 5
**Confidence:** 3

**Summary:**

This paper focuses on evaluating the performance of a supervised learning model $\theta: X \to Y$ over a set of subpopulations $Z \sim Q$. The authors consider a setting with "subpopulation" shift -- i.e., where the training data under/over samples data from a particular group. In this setting, they propose to bound the ``worst-case" group performance defined as: $$W_{\alpha}(\theta) := \max_{Q \in \mathcal{Q}_alpha} \mathbb{E}_Q[l(\theta(X),Y|Z)].$$ The authors present a new technique to estimate this quantity, and characterize its convergence in finite-sample settings. The authors apply their measures to determine the "worst-case" group-specific performance of models in 2 real-world prediction tasks: Warfarin dosage prediction, and land use from satellite images.



**Limitations And Societal Impact:**

Yes.

**Main Review:**

My overall impression is that this is promising work, but that it that requires some additional development to have long-term impact. The two most pressing weaknesses that I believe should be addressed include:

1. **Guidance for General-Purpose Applications**: The proposed performance evaluation framework requires that domain experts make difficult decisions – e.g., setting the value or $\alpha$ or the maximum level of acceptable loss. The paper would be stronger if the it included guidance for how to set these values. In particular, we need worked-out examples so that non-experts can use choose these values in an informed manner – ideally in Section 2 (Approach) and Section 4 (Experiments). The experiments in Section 4 focus on real-world datasets, but lack the type of guidance that would help real-world practitioners.

Ideally, I would recommend the authors provide this guidance for "common" machine learning applications. For example, if are using a binary classification task, then I would consider a setting where you are building a logistic regression model and where you measure performance in terms of AUROC or Expected Calibration Error (c.f. Squared Loss). These choices reflect the most common standards in clinical prediction models.

One related issue is to adapt the existing experiments to settings that are not "vacuous."For example, the "worst-case" 0-1 loss for all models in Table 1 currently exceed 50%. This result, which arises in the "worst-case," means that for subgroups larger than 1\% we might be better off using a coin-flip. The result is not wrong, but rather irrelevant. In a setting like this, we would ideally present practitioners with a meaningful guarantee (e.g., for $\alpha = 5%$) and a result stating the range of $alpha$ for which results are non-trivial.

2. **Comparisons to Simple Alternatives**: I enjoyed the fact that the current work focuses on the less "ambitious" scope of model evaluation. That being said, the paper needs to motivate and illustrate why we would want to use this machinery for evaluation as compared to existing alternatives to evaluate model performance. This could include: CV / Bootstrap / Parametric Bounds and the performance measurement approaches of Kearns/Herbert-Johnson.

**Time Spent Reviewing:**

3

---

> ### Author Response · Authors · 2021-08-10
> **Response to Reviewer 3a1D**
>
> We thank the reviewer for the thoughtful comments.
>
> ### Guidance on the choice of $\alpha$ and maximum acceptable loss
> We appreciate your raising this excellent point. As outlined in the central response above, we agree that this is an important problem and will provide detailed guidance for practitioners in the final version. We would like to emphasize two salient features of our worst-case formulation that are crucial to making an informed choice of $\alpha$ or the maximum acceptable loss: 1) the choice of attributes $Z$ can be arbitrary and flexible, allowing us to automatically consider complex intersectionality, and 2) the subpopulation size, $\alpha$, has a precise interpretation. Below, we expand on our central response and detail our plan for providing guidance on this important modeling choice.
>
> The choice of $\alpha$ should be informed by domain knowledge, as in all modeling tasks. First, the analyst should formulate whether $\alpha$ = 5%, 10%, 20%, 30% have meaningful ramifications in discussions with various stakeholders (e.g., product managers, customer focus groups, regulators, business executives). Second, we will provide useful heuristics for selecting $\alpha$ based on the training data. To illustrate our proposal, consider a simple scenario where we wish to require that prediction models perform uniformly well at least across racial groups and income. In this case, one should consider the intersection of race and low/middle/high income groups, and measure their proportion in the training data. By letting $(\mbox{race}, \mbox{income}) \subset Z$ and choosing $\alpha$ to be the proportion of the smallest such group, our worst-case subpopulation procedure ensures good performance over *all* subpopulations of similar size.
>
> As we wrote in the central response, it may be appropriate to evaluate the threshold $\alpha^\star$ where the model performance crosses the maximum level of acceptable loss. The maximum acceptable loss is often linked to a downstream outcome such as a health outcome, throughput of a system, or revenue. We anticipate this approach to be particularly useful in classification scenarios, where the loss (classification error, AUC etc) has a direct interpretation. Following the reviewer’s input, we will illustrate this in a binary classification problem, where practitioners often have desired levels of AUROC or calibration error. Towards illustrating this choice with real datasets, we have identified and begun preliminary data work on a recently constructed census dataset that includes an array of spatiotemporal distribution shifts.
>
> ### Clarification on FMoW experiments
> The worst-case subpopulation performance $W_{0.01}$ presented in Table 1 reports classification error across **62 classes**. An error rate larger than 50% is still substantially better than a random classifier. Nevertheless, we agree that $\alpha$ = 1% may be an overly small subpopulation size. Towards informing the conservative nature of our bounds, we already have evaluation results for a range of subpopulation sizes, and will make sure to include them in the final version.
>
> ### Comparison with standard model evaluation methods
> We would like to note that the de facto standard evaluation method is to calculate the out-of-sample average-case performance. As our worst-case subpopulation performance reduces to the average-case when $\alpha$ = 100%, we have included these results in both of our experiments. As such, Figures 2 and 4 can be viewed as already comparing against the de facto standard practice.
>
> In our final version, we will complement our existing analysis with a more thorough comparison of the statistical aspects of our approach. The confidence intervals on the performance of a model should grow wider as $\alpha$ becomes smaller, and we will compare our approach with confidence intervals for the average-case performance generated via a standard normal approximation or the bootstrap method as suggested by the reviewer. (Please also see our response to Reviewer e2UX for details on how to generate such confidence intervals.)
>
> ### Comparison with other fairness notions
> As we outlined in Lines 110-121 of our paper, our notion of worst-case subpopulation performance implements a Rawlsian notion of justice. By contrast, the fairness literature has largely focused on equalizing performance across demographic groups, viewing them as hard constraints. Previously proposed notions of subgroup fairness focuses on *equalizing* a notion of performance across subpopulations. Hebert-Johnson et al. [1] focus on equalizing calibration error across a *finite* number of subgroups. Kearns et al. [2, 3] consider equalizing notions of predictive performance (equality of opportunity or statistical parity) across exponentially many subgroups.
>
> Instead of requiring similar performance across groups (however bad they may be), we are interested in evaluating the performance on worst-off groups; we believe this is a practical and operationalizable notion of robustness across healthcare, tech products, business operations, and engineering applications such as robotics and autonomous vehicles. With the extra page afforded to us in the camera-ready version, we will expand our discussion on these connections and clarify our comparisons to existing notions of subgroup fairness. We remark that we are careful not to motivate our framework as driven by “fairness” considerations, as our perspective is more operational: we wish to ensure good performance almost all the time, by evaluating models across an infinite number of subpopulations.
>
> ### References
>
> [1] Hébert-Johnson, M. P. Kim, O. Reingold, and G. N. Rothblum. Calibration for the (computationally-identifiable) masses. arXiv:1711.08513 [cs.LG], 2017.
>
> [2] M. Kearns, S. Neel, A. Roth, and Z. S. Wu. Preventing fairness gerrymandering: Auditing and learning for subgroup fairness. arXiv:1711.05144 [cs.LG], 2018.
>
> [3] M. Kearns, S. Neel, A. Roth, and Z. S. Wu. An empirical study of rich subgroup fairness for machine learning. In Proceedings of the Conference on Fairness, Accountability, and Transparency, pages 100–109. ACM, 2019.

---

> > ### Comment · Reviewer_3a1D · 2021-09-01
> > **Thank you**
> >
> > Thank you for the response and the clarifications.

---

### Official Review · Reviewer_e2UX · 2021-07-23

**Rating:** 5
**Confidence:** 4

**Summary:**

This focus of this paper is a method for the evaluation of Machine Learning models with respect to their worst-case performance over all sufficiently large sub-populations. This has important applications in validating the performance of a model from a fairness perspective and ensuring that its predictions do not adversely affect a disadvantaged group.
The first contribution is to formalise the notion of “performance under worst-case subpopulation”. Here we have a distribution $P$ over random triples $(X,Y,Z)$ where $X$ is a covariate in $\mathcal{X}$, $Y$ is a label in $\mathcal{Y}$ and $Z$ corresponds to a set of protected variables e.g. income, ethnicity, gender etc. Let  $P_Z$ denote the associated marginal distribution with respect to a covariate $Z$. Given a lower-bound on the population $a \in (0,1]$, the associated collection of sub-populations is then given by,
$$\mathcal{Q}_a:=\left\lbrace Q_Z:P_Z= \tilde{a} \cdot Q_Z +(1-\tilde{a}) \cdot Q_Z' \text{ for some }a\geq {a} \text{ and  probability measure }Q_Z'  \right\rbrace$$. Given a predictive model $\theta: \mathcal{Z} \rightarrow \mathcal{Y}$ and a loss function $\ell:\mathcal{Y}^2 \rightarrow [0,\infty)$, the worst-case sub-population performance with lower bound $a$ is then defined by

$$ W_a(\theta):=sup_{Q_Z \sim \mathcal{Q}_{a}} E[E[\ell(\theta(X),Y)| Z] ] $$

where $Z$ is sampled from $Q_Z$ and $(X,Y)$ is sampled from the conditional distribution of $P$ given $Z$. The next goal is to estimate the worst-case sub-population performance $W_a(\theta)$ based on a data sample.

The second contribution of the paper is to develop an estimator $\hat{W}_a(\theta)$. In essence this proceeds by finding an estimate $\hat{h}$ for the conditional loss $h^*(Z):=E[\ell(\theta(X),Y)| Z]$, and then applying a variational formula.

The third contribution is to provide a finite sample guarantee (Theorem 1) which gives a high-probability bound between  $\hat{W}_a(\theta)$ and $W_a(\theta)$. This first result uses two assumptions: 1) A bounded loss function and 2) the conditional expectation  $h^*(Z):=E[\ell(\theta(X),Y)| Z]$ belongs to a known low-complexity class $\mathcal{H}$. The deviation bound is given in terms of local Rademacher complexity.

The fourth intended contribution is Theorem 2 which attempts a to give a tight deviation bound on  $\hat{W}_a(\theta)-{W}_a(\theta)$ without relying upon the complexity of the class $\mathcal{H}$..

Up to this point the lower bound $a$ is effectively viewed as a user-specified parameter. Next the authors turn to an alternative framework where the quantity of interest is not  ${W}_a(\theta)$ for a single value of $a$, but rather $a^*$, the smallest value $a$ for which $ {W}_a(\theta)$ is below a user specified upper bound on the loss. The final theoretical contribution is Theorem 3 which gives an estimator $\hat{a}$ for $a^*$ and a high-probability upper bound on the deviation between $\hat{a}$ and $a^*$.
The paper also includes an empirical illustration of the estimator in practice on two data sets for which worst-case performance is of interest.


**Ethical Concerns:**

No ethical concerns.

**Limitations And Societal Impact:**

As discussed above, the work is promising with a potential for positive societal impact in providing fairer applications of ML.

There are a few limitations to the present evaluation, also discussed above. One issue with the paper is that there is little discussion of the optimality of the bounds. It seems very likely that the $n^{-1/2}$ rate in Theorem 1 is optimal. The optimality of Theorem 3 is much less clear.


**Main Review:**

Providing Machine Learning methods which perform well for across diverse sub-populations without adversely affect a disadvantaged group is an crucial challenge of great importance to the Machine Learning community, and a potential for significant societal impact. Here, the goal is slightly more modest: The intention is to evaluate the worst-case performance of an existing trained model. Nonetheless, this can be viewed as building block towards learning models which perform well across diverse sub-populations. In this sense the problem is a significant one.

An interesting connection is drawn with the Rawlsian approach to social justice in which social inequalities can only be justified if they improve the situation for the worst-off within society. The direct ML analogue here being that we can justify a heterogenous performance of a predictive model only in so far as this results in a better performance for all sub-populations.

As noted by the authors, the problem has a great deal in common with the recent work of Duchi et al. (2020) [33] on distributionally robust learning which focuses on the more challenging problem of learning rather than evaluation. In light of Duchi et al’s prior the present paper is not quite so original from a conceptual perspective. Nonetheless, the problem of evaluation for existing models is arguably of interest in its own right.

In terms of quality the paper has potential, but a few significant issues. Let’s address each of the theorems in turn before turning to the experiments.

Theorem 1 is the strongest result in the paper. An high-probability guarantee is provided for the deviation between $\hat{W}_a(\theta)$ and $W_a(\theta)$. The presence of the square root of the local Rademacher complexity (rather than the standard Rademacher complexity) is of interest. This is a consequence of the difficulty of estimating $h^*(Z):=E[\ell(\theta(X),Y)| Z]$ by ERM on the squared loss, with the square root (and the parametric rate of $n^{-1/2}$ originating from the move from L1 to L2 error via Cauchy-Schwarz (see line 731). The proof proceeds by combining existing results on together in an interesting way. One major limitation of the result is the use of Assumption 2 ie. The assumption that $h^*(Z):=E[\ell(\theta(X),Y)| Z]$ belongs to a pre-specified low-complexity class. Here an oracle type bound would be preferable. That is, rather than include Assumption 2 (which is less interpretable than the corresponding realisability assumption in say classification), the authors could include an approximation error term so that the bound depends upon the deviation between the conditional expectation $E[\ell(\theta(X),Y)| Z]$  and the best approximator in the class.
There are a couple of minor issues surrounding the presentation of Theorem 1. Firstly, it would be good to reference Mendelson near the statement of Lemma 2 (i.e. within the main paper rather than the Appendix). As another minor point, the proof of Theorem 1 seems slightly over-complicated since Assumption 1 asserts boundedness. Hence, the full machinery of Orlicz norms is not really necessary.

In the case of Theorem 2 it would be very useful to include a discussion when we should expect the quantity $\Delta_n$ to be small. Note that in general we would not expect this quantity to converge to zero as $n\rightarrow \infty$.

Theorem 3 requires further explanation. In particular, it seems that the denominator on the right hand side of the inequality could be arbitrarily close to zero. Indeed, for the bound to be meaningful we require $a^*\geq \underline{a}$ which suggests that the expectation in the denominator is low. Some discussion of optimality would be very helpful here.

There are also some limitations regarding the experimental section. This section demonstrates some potential applications of the method. However, it would be more interesting to see an empirical demonstration of the efficacy of the method. This could be done by evaluating on synthetic data, where the estimated quantity is known. Another potential issue is the confidence intervals in Figure 2. How should we interpret these? If they originate in the bound in Theorem 1 it seems strange that they are not much larger for the more flexible model classes? It would also be good to include confidence intervals in Figure 3.

Overall, I found the idea to be interesting, with significant promise and potential for valuable impact.

My remaining concerns with regards to this paper concern the empirical results section and the degree of optimality in the theoretical results. Please see below for more details.

Updates following author response:

Thanks for your careful response which provided a great deal of clarity. Please see below for more details.

1)	Misspecification error in Theorem 1

Thanks for the clarification. I believe dropping Assumption 2 and including this additional term would be a small change that improve the result.

2)	Incorporating heavier-tailed losses

It would indeed be interesting to include these results in the main the text. An alternative would be to adhere to the current boundedness assumption and give the simpler version of the proof accordingly.

3)	Correctness of Theorem 2

Yes! My mistake entirely – Apologies. The proof is indeed correct. Thank you for the clarification.

Here I think its important to underline that we can’t in general expect $\Delta_n$ to be small even if the model class is arbitrarily rich, and the sample size is arbitrarily large. Indeed, we can would only expect this to hold if $l(\theta(X),Y)$ has zero conditional variance given $Z$.

4)	Discussion on Theorem 3

Thanks for the clarification. The key point here is that there are natural situations where the bound in Theorem 3 is uninformative. For example, we are particularly interested in situations where $\alpha_*$ is small (i.e. the learnt function is fair). However, for sufficiently small $\alpha_*$, the $1-\alpha_*$-quantile will typically be very close (or even equal) to the essential supremum. Whenever this occurs the RHS of the bound will blow-up. Perhaps this is a necessary feature of any bound of this form.

5)	Confidence intervals and experiments

The main result of the paper (Theorem 1) yields confidence intervals for $W_\alpha$. As such, it is somewhat surprising that these are not used in the simulations (instead of 5 random repeats).

I also believe it’s strange that only Figure 2 includes confidence intervals (i.e. not Figure 4). In the final version we should see confidence intervals for all experiments.

Overall, with these experiments it remains unclear what conclusions we can draw regarding the efficacy of the method since there is no method-independent standard of correctness. For this reason I would suggest giving greater emphasis to experimental results on synthetic data where the quantity being estimated is known a priori.

The CLT-type asymptotically valid confidence intervals would certainly make for an interesting development. Of course it may be difficult to include these results whilst also providing further clarification on the existing results.


6)	Optimality

In each of the theoretical results – Theorems 1,2,3 a discussion of optimality would greatly improve the paper. In the case of Theorem 1 it seems highly plausible that the $O(n^{-1/2})$ rate is optimal. Still, it would be useful to provide some argument for this in the text. In the case of Theorem 2 the $O(n^{-1/4})$ is unusual and some discussion of optimality would be especially valuable. Similarly, the denominator in the bound of Theorem 3 can be expected to be very small under natural settings, and again a discussion of optimality would be really valuable here.


**Time Spent Reviewing:**

8

---

> ### Author Response · Authors · 2021-08-10
> **Response to Reviewer e2UX**
>
> Thank you for the detailed review. We will incorporate all the expository suggestions in our final version. Please see our main response above for an overview of our primary motivation and comparisons to Duchi et al. (2020).
>
> ### Misspecification error in Theorem 1
> Thank you for the helpful comments on this result. When there is model misspecification (i.e., if Assumption 2 doesn’t hold), we can still prove a variant of Theorem 1. We will update our statement of Theorem 1 to incorporate this flexibility. We have the same bound as before, but with an additional term representing misspecification error. More formally, if we denote the best possible estimator in the model class by $\tilde{h} := \arg\min_{h\in\mathcal{H}} \mathbb{E} (\ell(\theta(X);Y) - h(Z))^2$, then the RHS of Lemma 8 would have an extra term of $\mathbb{E}(\tilde{h} - h^\star)^2$, which measures the misspecification error of the model class and, in particular, equals zero under well-specification. The local Rademacher complexity we used for $r^\star$ would be centered around $\tilde{h}$ instead. All other parts of the proof continue to hold.
>
> ### Incorporating heavier-tailed losses
> In regards to Appendix Section A.1, we showed Proposition 4 assuming sub-Gaussianity rather than boundedness of the loss function, in expectation of deriving a more general finite-sample concentration guarantee in later versions. After our initial submission, we have proved finite-sample concentration guarantees for heavy-tailed loss functions, including sub-Gaussian random variables. Our new result builds on Mendelson (2014)’s small ball method [1], and improves our existing Theorem 1 even when losses are bounded. We will present these more general results in the final version of the paper, towards considering heavier-tailed losses.
>
> ### Correctness of Theorem 2
> Our proof for Theorem 2 is correct, although we could have done a better job writing out the important steps more explicitly. In showing inequality (17), we use the fact that $\text{LHS}-\text{RHS} = -|S_2|^{-1}\sum_{i\in S_2}\zeta_i^2 \le 0$. Please note the magnitude of this difference is what we anticipate $\Delta_n$ to be close to for well-specified or rich enough model classes. We thank the reviewer for raising this point, and we will make sure to clarify this in our subsequent revision.
>
> ### Discussion on Theorem 3
> We appreciate the comments on Theorem 3. The theorem states that for any $\underline{\alpha}$, we know either the true $\alpha^\star$ is sufficiently small ($\alpha^\star < \underline{\alpha}$), indicating the model $\theta$ meets the robustness test, or it is close to our estimator $\widehat{\alpha}$. Since the threshold $\underline{\alpha}$ is arbitrary and the denominator in the inequality decreases as $\underline{\alpha}$ decreases, the theorem presents a tradeoff in setting $\underline{\alpha}$ between the level of robustness and the accuracy of our estimator $\widehat{\alpha}$. We will make sure to include a more thorough discussion of this result.
>
> ### Confidence intervals and experiments
> In light of our primary motivation described above, we decided to focus on real-use cases of our methodology given the limited space. With the extra page afforded to us in the camera-ready version, we will make sure to include additional simulation examples that verify the statistical convergence of our estimators. We would like to clarify that the error bars in Figure 1 are based on 5 different seeds for each method, which is why they do not vary with the size of the model class. While this is standard empirical practice, our goal is to provide statistical methods that can generate asymptotically calibrated confidence intervals for the worst-case subpopulation performance. This is a technically involved problem due to the two-stage nature of our procedure. Building on the literature on semiparametric inference, we have preliminary analysis proving a central limit result for our procedure: denoting by $\hat{h}$ our estimator of $h^*$, if $\mathbb{E}(\hat{h}(Z) - h^*(Z))^2$ converges at $o_p(n^{-½})$ rates, we have shown the central limit result $\sqrt{n} (\hat{\omega}_{\alpha} - \omega_\alpha) \Rightarrow N(0,\sigma^2)$.
>
> In the final version of our paper, we will 1) include a complete asymptotic theory for generating confidence intervals for the worst-case subpopulation performance with exact asymptotic coverage, 2) perform a thorough evaluation of the statistical methodology presented in this work, and 3) include valid confidence intervals for all of our experiments.
>
> Once again, we are grateful for the reviewer’s detailed suggestions.
>
> ### References
> [1] S. Mendelson. Learning without Concentration. arXiv:1401.0304 [cs.LG], 2014.

---

> ### Author Response · Authors · 2021-08-18
> **Follow-up response to Reviewer e2UX (1)**
>
> Thank you for the updated review. We appreciate your close reading and prompt feedback. Below, we provide further responses.
>
> ### Theorem 2
>
> > Here I think it’s important to underline that we can’t in general expect $\Delta_n$ to be small even if the model class is arbitrarily rich, and the sample size is arbitrarily large. Indeed, we can would only expect this to hold if $\ell(\theta(X);Y)$ has zero conditional variance given $Z$.
>
> We fully agree&mdash;thank you for bringing this up. We will make sure to explicitly mention this in the camera-ready version. We would like to emphasize that while we cannot expect $\Delta_n$ to converge to zero as $n\to\infty$, it may still be small enough to provide informative bounds for any finite $n$ and high-dimensional feature vector $Z$, even when standard Rademacher-based guarantees (Theorem 1) provide vacuous generalization bounds. When using deep networks to predict the conditional risk $h^\star(Z) := \mathbb{E}[\ell(\theta(X);Y)|Z]$, typical generalization bounds such as Theorem 1 that are based on the Rademacher complexity become vacuous due to the high-dimensional nature [1, 2]. In these scenarios, Theorem 2 provides an attractive *dimension-free* alternative, as the out-of-sample error $\Delta_n$ is typically small for rich model classes such as neural networks.
>
> Consider, for example, ImageNet-scale settings where we wish to guarantee robustness over distribution shifts in the image distribution. Models typically have hundreds of millions of parameters (e.g. ResNet, vision transformers) but the dataset size is in the order of 1-2 million. Our Theorem 1 will provide vacuous generalization guarantees in such high-dimensional scenarios [1, 2], yet Theorem 2 will still give useful bounds as modern architectures achieve small out-of-sample error $\Delta_n$.
>
> ### Theorem 3
>
> > Thanks for the clarification. The key point here is that there are natural situations where the bound in Theorem 3 is uninformative. For example, we are particularly interested in situations where $\alpha^*$ is small (i.e. the learnt function is fair). However, for sufficiently small $\alpha^*$, the $1-\alpha^*$-quantile will typically be very close (or even equal) to the essential supremum. Whenever this occurs the RHS of the bound will blow-up. Perhaps this is a necessary feature of any bound of this form.
>
> Although Theorem 3 provides a concentration guarantee on the estimator $\widehat{\alpha}$, it is fundamentally a *robustness test*. This means that even though the high-probability bound is loose when $\alpha^*$ is small, the theorem is able to recognize this fact through the alternative conclusion of $\alpha^*<\underline{\alpha}$. We are thus able to *certify* that the model is indeed robust enough to subpopulation shifts.
>
> We agree with your comment on the inevitable looseness in the corner case of very small $\alpha$’s and the need for an increasing number of samples to ensure that relevant guarantees continue to hold non-trivially in such cases. This difficulty is fundamental [3] as our worst-case subpopulation performance only focuses on $\alpha$-fraction of the data (see below for more discussion). We will make sure to explicitly note this in the camera-ready version.
>
> ### Experiments
> > The main result of the paper (Theorem 1) yields confidence intervals for $W_\alpha$. As such, it is somewhat surprising that these are not used in the simulations (instead of 5 random repeats). I also believe it’s strange that only Figure 2 includes confidence intervals (i.e. not Figure 4). In the final version we should see confidence intervals for all experiments.
>
> Theorem 1 is a finite-sample high-probability guarantee, given as inequalities, on our estimator $\widehat W_\alpha (\widehat h)$ of the worst-case subpopulation performance $W_\alpha$. The cost of such a finite-sample result is that the confidence interval it generates is often loose due to large universal constants. In contrast, asymptotic results such as the central limit theorem provide confidence intervals $U_n$ with asymptotically exact coverage probability. For example, if the desired confidence level is 95%, a central limit result allows constructing $U_n$ such that $\lim_{n \to \infty} \mathbb{P}( \mbox{W}_{\rm \alpha} \le U_n) = 0.95$. (Analogously, applied statisticians rarely calibrate confidence intervals based on finite-sample Hoeffding-type bounds, and instead almost always rely on the central limit theorem.)
>
> This is precisely the reason we have undertaken a serious effort to develop asymptotic guarantees for our procedure, so that we can obtain confidence intervals with asymptotically exact coverage probability. (Please see our previous response.)  In our initial submission, we nevertheless wanted to follow standard empirical best practices in ML and generate heuristic error bars when possible. We were constrained by our academic computing budget in our final large-scale experiment (Figure 4). In the final version of the paper, we will make sure to include asymptotically calibrated confidence intervals for all experiments, in addition to a thorough simulation study analyzing the coverage probabilities of our principled confidence intervals.
>
> > Overall, with these experiments it remains unclear what conclusions we can draw regarding the efficacy of the method since there is no method-independent standard of correctness. For this reason I would suggest giving greater emphasis to experimental results on synthetic data where the quantity being estimated is known a priori.
>
> Thank you for raising this point. We agree that a simulated experiment can illustrate the statistical properties of our method; we have begun running such experiments and will make sure to include them in the camera-ready version. We would like to emphasize that given the array of theoretical results that verify the statistical validity of our approach, in our initial submission we had chosen to use the limited space to showcase the practical use-cases of our method (as they complement each other). In particular, our existing experiments demonstrate the scalability of our procedure (even on large-scale models such as CLIP). In Table 1, the CLIP model exhibits worse average-case performance than other SOTA models, but its robustness properties as claimed in [4] is evident from our proposed approach: it matches or outperforms other SOTA models in terms of the worst-case subpopulation performance over (label, region, year).
>
> (To be continued)

---

> ### Author Response · Authors · 2021-08-18
> **Follow-up response to Reviewer e2UX (2)**
>
> (Continued)
>
> ### Optimality
>
> > In each of the theoretical results – Theorems 1,2,3 a discussion of optimality would greatly improve the paper. In the case of Theorem 1 it seems highly plausible that the $O(n^{-1/2})$ rate is optimal. Still, it would be useful to provide some argument for this in the text. In the case of Theorem 2 the $O(n^{-1/4})$ is unusual and some discussion of optimality would be especially valuable. Similarly, the denominator in the bound of Theorem 3 can be expected to be very small under natural settings, and again a discussion of optimality would be really valuable here.
>
> Instead of the typical asymptotic perspective, we are considering the high-dimensional nature of modern learning applications: the model class $\mathcal{H}$ typically grows richer with sample size and the constant in the typical $O(n^{-1/2})$ rate can dominate the generalization bound. The rate in Theorem 1 is optimal in the large-sample limit, but it may be prohibitively loose in high-dimensional learning scenarios. On the other hand, the out-of-sample error $\Delta_n$ in Theorem 2 can often be made small by rich model classes along with ML best practices developed in the last two decades: cross-validation, versatile neural network architectures such as ResNets and transformers, and feature engineering. This guarantee is thus fundamentally different from traditional approaches that use uniform concentration inequalities (Theorem 1) and embodies our ethos of developing scalable and practical evaluation methods with provable guarantees.
>
> Further note that the rate in Theorem 2 is not $O(n^{-1/4})$ in the large sample limit&mdash;it is constant as $\Delta_n$ may not go to zero when $n \rightarrow \infty$. Nevertheless, it is the first known *dimension-free* result of its kind, making it scalable to extremely large-scale modern ML models whose complexity (the number of parameters for over-parameterized models like deep networks) greatly outnumbers the sizes of data sets $n$.
>
> For any small $\alpha$, the minimax estimation error for the worst-case subpopulation performance should scale as $\Theta( (\alpha \cdot n)^{-1/2})$ since we are effectively only using a $\alpha$-fraction of the data. We believe that a minor modification of [3] should give a information theoretic minimax lower bound of this form. However, we would like to note that the main goal of this work is to develop practical and scalable evaluation procedures. Minimax optimality of our convergence results, while theoretically appealing, often measures performance against an overly conservative data-generating distribution. We hence view a full characterization of information-theoretic lower bounds outside the scope of this work, but fully agree that these are nevertheless important intellectual directions. We very much appreciate the detailed input, and will make sure to explicitly discuss these directions as future work in the paper.
>
>
> ### References
> [1] C. Zhang, S. Bengio, M. Hardt, B. Recht, and O. Vinyals. Understanding deep learning requires rethinking generalization. In *Proceedings of the Fifth International Conference on Learning Representations*, 2017.
>
> [2] P. Bartlett, D. J. Foster, and M. Telgarsky. "Spectrally-normalized margin bounds for neural networks." arXiv preprint arXiv:1706.08498, 2017.
>
> [3] D. Levy, Y. Carmon, J. C. Duchi, and A. Sidford. Large-Scale Methods for Distributionally Robust Optimization. arXiv preprint arXiv:2010.05893, 2020.
>
> [4] A. Radford, J. W. Kim, C. Hallacy, A. Ramesh, G. Goh, S. Agarwal, G. Sastry, A. Askell, P. Mishkin, J. Clark, et al. Learning transferable visual models from natural language supervision. In *Proceedings of the 38th International Conference on Machine Learning*, 2021.

---

### Author Response · Authors · 2021-08-10
**Response to questions shared among multiple reviewers**

We thank the reviewers for the considered and helpful feedback. We truly appreciate the reviewers’ time and detailed input. Without mention, we will incorporate all feedback to improve our exposition. We first address the questions shared among multiple reviewers and then detail our response to individual questions as respective replies.

### Overview
Our work aims to provide a scalable yet rigorous methodology for evaluating the robustness of state-of-the-art prediction models. Recent advances in self-supervised learning [1, 2, 5, 6] allow training models on unprecedented scales, and engineered approaches have found wide success in applications such as computer vision and natural language processing. For example, the recent CLIP model from OpenAI [4] was trained on 400M image-text pairs; the image data alone is two orders of magnitude larger than ImageNet. Yet our theoretical understanding of training neural networks remains limited at best despite recent efforts, and generalization guarantees for deep networks are vacuous [6].

For these heuristics and engineered training approaches to bear fruit and to transform high-stakes applications, we need to rigorously evaluate models across a range of subpopulations. We develop methods for *provably* evaluating the worst-case subpopulation performance of a model. By focusing on the narrower question of model evaluation, we are able to develop estimation procedures with theoretical guarantees that apply to state-of-the-art ML settings (see our Theorem 2, as well as our discussion below.).

### Comparison to DRO methods
Our approach is in stark contrast to previous distributionally robust optimization methods that aim to train models with respect to a notion of worst-case performance. Most notably, Duchi et al. (2020) [3] recently proposed methods for optimizing the worst-case subpopulation performance. However, such optimization methods cannot scale to modern learning problems both computationally and statistically. Their training algorithm requires a full batch, prohibiting its use in even ImageNet-scale problems; each gradient step requires solving a convex program with $O(n^2)$ variables, where $n$ is the number of the data points. Statistically, the generalization error of their method scales as $O(n^{-1/d})$, exacerbating exponentially with the input dimension $d$; these errors become prohibitively large even when $d > 20$. By virtue of only focusing on model evaluation, we are able to evaluate state-of-the-art models (e.g., CLIP [4]) in problems that are 4-6 orders of magnitude larger. We thank the reviewers for raising this question, and will better illustrate these distinctions in our paper.

### Choice of $\alpha$ or the maximum level of acceptable loss
Selection of the subpopulation size $\alpha$ is a nontrivial yet important modeling choice. A crucial part of our formulation is that $\alpha$ has a concise and precise interpretation, which is critical to leveraging domain knowledge to inform its choice. As reviewers have pointed out, we will provide practical guidance on this choice using our real-world experiments as examples; thank you for raising this important issue.

First, we will provide heuristics for informing the choice of $\alpha$ using the size of proxy subgroups in the training data (e.g., Blacks with age > 60). Such selection of $\alpha$ provides strong guarantees over *all* subpopulations of a similar size.

Alternatively, it may be appropriate to evaluate the threshold $\alpha^\star$ where the model performance crosses the maximum acceptable loss. We anticipate this approach to be particularly useful in classification scenarios, where the loss (e.g., classification error / AUC) has a direct interpretation. As regression losses are more subtle, we will illustrate how to inform the choice of the maximum acceptable loss in the Warfarin dataset using domain knowledge on dosages.

### On the benefits of Theorem 2
Our evaluation algorithm enjoys data-dependent convergence guarantees (Theorem 2). This allows our method to scale to settings where existing generalization bounds are prohibitively loose [6], e.g., when using deep networks to predict the conditional risk $h^\star(Z) := \mathbb{E}[\ell(\theta(X);Y)|Z]$. In practice, the out-of-sample error $\Delta_n$ in our generalization bound can be made small by rich model classes along with ML best practices developed in the last two decades: cross-validation, versatile neural network architectures such as ResNets and transformers, and feature engineering. This guarantee is thus fundamentally different from traditional approaches that use uniform concentration inequalities (e.g., Theorem 1) and embodies our ethos of developing scalable and practical evaluation methods with provable guarantees. Instead of the typical asymptotic perspective, we are considering the high-dimensional nature of modern learning applications: the model class $\mathcal{H}$ typically grows richer with sample size, and we expect $\Delta_n$ to be small. In fact, the result is stronger than we had initially stated, holding under misspecified models for $h^\star(Z)$ (i.e., without requiring Assumption 2).

Regarding Reviewer e2UX’s comments, we would like to clarify that our existing proof is correct, although admittedly terse at times. In the inequality in Eq. (17), we used the fact that $\text{LHS}-\text{RHS} = -|S_2|^{-1}\sum_{i\in S_2}\zeta_i^2 \le 0$. We will make sure to expand and clarify this point in the proof of Theorem 2.

Thank you again for all the helpful comments. We have begun revising our manuscript and will incorporate every suggestion in this central response and individual replies below.

### References

[1] T. B. Brown, B. Mann, N. Ryder, M. Subbiah, J. Kaplan, P. Dhariwal, A. Neelakantan, P. Shyam, G. Sastry, and A. Askell. Language models are few-shot learners. arXiv:2005.14165 [cs.CL], 2020.

[2] T. Chen, S. Kornblith, M. Norouzi, and G. Hinton. A simple framework for contrastive learning of visual representations. Proceedings of the 37th International Conference on Machine Learning, 2020.

[3] J. Duchi, T. Hashimoto, and H. Namkoong. Distributionally robust losses for latent covariate mixtures. arXiv:2007.13982 [stat.ML], 2020.

[4] A. Radford, J. W. Kim, C. Hallacy, A. Ramesh, G. Goh, S. Agarwal, G. Sastry, A. Askell, P. Mishkin, J. Clark, et al. Learning transferable visual models from natural language supervision. In Proceedings of the 38th International Conference on Machine Learning, 2021.

[5] Q. Xie, M.-T. Luong, E. Hovy, and Q. V. Le. Self-training with noisy student improves imagenet classification. In Proceedings of the 30th IEEE Conference on Computer Vision and Pattern Recognition, pages 10687–10698, 2020.

[6] C. Zhang, S. Bengio, M. Hardt, B. Recht, and O. Vinyals. Understanding deep learning requires rethinking generalization. In Proceedings of the Fifth International Conference on Learning Representations, 2017.

---

### Decision · Program_Chairs · 2021-09-27

**Decision:**

Accept (Poster)

**Comment:**

The paper provides some interesting theoretical results on estimating the worst-case performance of a model over all subpopulations of a given size, and has supporting experimental evaluations. The reviews generated a detailed back-and-forth discussion with the authors. The main concerns raised by the reviewers are about the lack of sufficient experimental results demonstrating the efficacy of the proposed method and the lack of practical guidance for choice of the group size parameter $\alpha$.

My impression of the paper is that the results presented would be of interest to the ML fairness community, and so I would recommend accepting it. However, it's very important that the authors do a thorough job of incorporating the following changes in the final version of the paper (in addition to the other changes that they have promised to make in their response):
- Simulation study analyzing the asymptotic convergence of the proposed estimator
- Experiments illustrating how the parameter $\alpha$ should be set

I would also like to emphasize that the reviewers were chosen from diverse backgrounds within ML to assess both the theoretical and practical aspects of the work, and I think they have done a thorough job of reviewing the paper. I understand that there were some initial questions about the correctness of Theorem 2 (which, to be honest, I too had during my initial reading of the paper), and I am glad that the authors were able to satisfactory resolve them. I trust that the authors will use all the feedback provided and make all the changes they have promised in their response.